# The Illusion of Generalization in Tabular Language Models

**Aditya Gorla** [1]   **Ratish Puduppully** [2]

## Abstract

Tabular Language Models (TLMs) have been claimed to achieve strong generalization for tabular prediction. We conduct a systematic re-evaluation of Tabula-8B as a representative TLM, utilizing 165 datasets from the UniPredict benchmark. Our investigation reveals three findings. First, binary and categorical classification achieve near-zero median lift over majority-class baselines and strong aggregate performance is driven entirely by quartile classification tasks. Second, top-performing datasets exhibit pervasive contamination, including complete train-test overlap and task-level leakage that evades standard deduplication. Third, instruction-tuning without tabular exposure recovers 92.2% of standard classification performance and on quartile classification, format familiarity closes 71.3% of the gap with the residual attributable to contaminated datasets. These findings suggest claimed generalization likely reflects evaluation artifacts rather than learned tabular reasoning. We conclude with recommendations for strengthening TLM evaluation.[1]

## 1. Introduction

Large Language Models (LLMs) have transformed natural language processing (Zhao et al., 2025; Minaee et al., 2025), and their success has inspired a new wave of methods for tabular data (Fang et al., 2024; Gardner et al., 2024; Hegselmann et al., 2023). Beyond naive approaches such as text serialization (Dinh et al., 2022; Lee et al., 2025), a new class of language models specifically (pre-)trained or fine-tuned for tabular domain have emerged (Fang et al., 2024; Hegselmann et al., 2023; Gardner et al., 2024; Wang et al., 2023; Sun et al., 2024). We refer to these as *Tabular Language*

*Models* (TLMs). The putative hypothesis behind TLMs mirrors that of LLMs and Vision-Language Models (VLMs): with sufficient scale, these models may learn to generalize across the structure, invariances, and patterns inherent in tabular data, enabling zero- and few-shot prediction, imputation, and synthetic generation across binary, categorical, and continuous data types (Hegselmann et al., 2023; Wang et al., 2023; Gorla et al., 2025; Borisov et al., 2023).

Tabular data, however, possesses three properties that jointly distinguish it from text and images (Fang et al., 2024; Borisov et al., 2024; Grinsztajn et al., 2022). First, tabular data is *row-permutation invariant*: the ordering of samples carries no semantic meaning. Second, and more critically, tabular data is *column-permutation invariant*. Unlike text (which has sequential structure) or images (which have spatial locality), features in a table have no *a priori* or universal ordering[2]. Third, tabular data is *heterogeneous*, spanning binary, categorical, and continuous types. This is in stark contrast to text (discrete tokens from a fixed vocabulary) or images (bounded pixel intensities). One can reasonably argue these properties explain why gradient-boosted trees (GBTs) and Prior-Fitted Networks (PFNs) have excelled on tabular tasks (Grinsztajn et al., 2022; Shwartz-Ziv & Armon, 2022; Hollmann et al., 2023; 2025). GBTs naturally handle irregular, discontinuous patterns without assuming feature relationships (Chen & Guestrin, 2016; Grinsztajn et al., 2022), while PFNs encode inductive biases through synthetic (heterogeneous) data priors and leverage transformer architectures without fixed positional encodings to respect permutation invariance (Hollmann et al., 2023; Müller et al., 2022).

Against this backdrop, recent work on TLMs claims significant zero- *and* few-shot improvements over both GBTs and PFNs (see Section 2) (Gardner et al., 2024; Sun et al., 2024; Hegselmann et al., 2023). This finding, while exciting, is surprising upon closer examination. TLMs do not explicitly encode tabular-specific inductive biases. Instead they serialize tables into sequential text, tokenize heterogeneous values into approximate linguistic tokens, and inherit priors from next-token prediction that assume sequential structure. In principle, transformers can learn arbitrary de-

---

[1]University of California, Los Angeles, USA [2]IT University of Copenhagen, Denmark. Correspondence to: Aditya Gorla <adityagorla@ucla.edu>, Ratish Puduppully <rapu@itu.dk>.

*Proceedings of the $43^{rd}$ International Conference on Machine Learning*, Seoul, South Korea. PMLR 306, 2026. Copyright 2026 by the author(s).

[1]Code and artifacts are available at https://github.com/ratishsp/tlm-illusion.

---

[2]While domain experts may arrange features meaningfully, this ordering is not intrinsic to the data semantics

pendencies regardless of input format, as demonstrated by Vision Transformers overcoming the lack of convolutional priors (Dosovitskiy et al., 2021). Yet it remains unclear whether TLMs *actually* learn tabular invariances and generalize over tabular data or succeed through other mechanisms entirely. This tension between empirical claims and architectural considerations motivates our central question: **Do TLMs truly *generalize* to tabular data? And if so, why?**

Our skepticism is warranted by a growing body of work scrutinizing common claims about foundation models more broadly. Rogers & Luccioni (2024) argue that LLM evaluation is often confounded by benchmark contamination, inflated claims of emergent abilities, and insufficient attention to what models actually learn from their training data.

Before investigating *why* TLMs might generalize, we must first verify *that* they do. Following Lipton & Steinhardt (2019), we conduct a systematic empirical investigation to "consider and rule out alternative hypotheses."

**Contributions.** Our investigation uncovers three findings that suggest TLMs' reported *performance gains likely **do not** reflect genuine tabular generalization*:

- **Task-type bias:** Decomposing performance by task type reveals that TLMs excel primarily on continuous (quartile) regression while underperforming simple baselines on binary classification.
- **Data contamination:** We present systematic evidence of training-test overlap and direct leakage in standard benchmarks.
- **Format over structure:** Instruction-tuning alone achieves competitive performance, suggesting TLMs may leverage instruction-following capabilities rather than learning tabular-specific representations.[3]

Based on these findings, we conclude with 7 recommendations for strengthening evaluation standards to differentiate evaluation artifacts from substantive advances in TLM development.

## 2. Background

**Tabular Prediction Methods.** Gradient-boosted trees (GBTs) have dominated tabular prediction for over two decades (Friedman, 2001; Chen & Guestrin, 2016; Ke et al., 2017; Prokhorenkova et al., 2018). Recent efforts to close the gap between deep learning and GBTs include Tab-Net (Arik & Pfister, 2021) and FT-Transformer (Gorishniy et al., 2021), yet comprehensive benchmarks show that neural approaches still struggle to consistently outperform well-tuned GBTs (Grinsztajn et al., 2022; Shwartz-Ziv &

Armon, 2022). This gap has been attributed to differing inductive biases (see above Section 1) (Grinsztajn et al., 2022). Alternatively, TabPFN (Hollmann et al., 2023) proposes training a transformer on synthetic data to perform in-context learning on small tabular datasets (Hollmann et al., 2025). While PFNs achieve strong performance in small-data regimes, they are designed for small datasets and rely on synthetic rather than real-world pretraining data.

**Tabular Language Models.** TLMs fine-tune or pretrain LLMs on serialized tabular data, converting rows into text sequences via templates or special tokens. TabLLM (Hegselmann et al., 2023) demonstrated few-shot classification by serializing rows and prompting frozen LLMs. UniPredict (Wang et al., 2023) scaled this approach to 169 Kaggle datasets, claiming universal classification capabilities. More recent work has explored scaling laws for tabular language modeling (Sun et al., 2024) and generation of synthetic tabular data (Borisov et al., 2023). These methods commonly report zero- and few-shot improvements over GBTs and TabPFN on aggregate accuracy metrics.

However, TLM evaluations rarely include majority-class baselines or stratify results by task type. We surveyed five recent TLM papers spanning major venues and found that *none* report comparisons against majority-class baselines (Hegselmann et al., 2023; Wang et al., 2023; Sun et al., 2024; Kim et al., 2024; Fang et al., 2024) (Appendix G). This omission is particularly consequential for tabular benchmarks, which frequently contain imbalanced datasets from domains such as fraud detection, medical diagnosis, and customer churn (Borisov et al., 2024). Without baseline comparison, it is impossible to distinguish genuine learning from exploitation of class frequencies. Chance-corrected metrics such as Cohen's Kappa (Cohen, 1960) are similarly absent from the TLM literature, despite being standard practice in classification evaluation (Japkowicz & Shah, 2011).

As our case study, we focus on Tabula-8B (Gardner et al., 2024), which represents the current frontier of open, large-scale TLMs. Tabula-8B fine-tunes Llama 3-8B (Grattafiori et al., 2024) for classification and binned regression using a language modeling objective. It is trained on T4, a filtered subset of TabLib (Eggert et al., 2023) containing approximately 4 million tables (~100B tokens) extracted from Common Crawl and GitHub. The authors evaluate on 329 datasets across five benchmarks, reporting 5–15 percentage point gains over XGBoost and TabPFN in few-shot settings.

**Contamination Concerns in Foundation Models.** Data contamination, the presence of evaluation data in training corpora, is a recognized concern in foundation model. Brown et al. (2020) with GPT-3 were among the first to systematically analyze contamination, using $n$-gram overlap detection and finding variable impact across bench-

---

[3]We present this as a hypothesis our evidence supports, while acknowledging that the boundary between "format learning" and genuine "understanding" remains difficult to delineate precisely.

marks. More recently, Bordt et al. (2024) demonstrated that GPT-3.5 and GPT-4 have memorized many popular tabular datasets verbatim, including the entire Iris and Wine datasets from UCI. They show that memorization leads to inflated performance estimates, with accuracy dropping by 6 percentage points when datasets are perturbed to break memorization cues. This finding directly parallels our observations in TLM evaluation.

Gardner et al. (2024) acknowledge that "at most one-third of benchmark tables may occur at least once" in T4, but claim minimal impact citing prior work on vision-language models (Radford et al., 2021; 2019). However, tabular data poses unique contamination challenges. Standard row-level deduplication is insufficient since the same records can appear with (slightly) different column names across dataset versions, identical data can be replicated across multiple tables with varying schemas, and task-level associations may enable solving evaluation tasks even when specific records are absent (also see Section 4.2.3). During the preparation of this manuscript, we found concurrent work (Joshi et al., 2026) that independently surfaced similar concerns in the VLM domain. Joshi et al. (2026) found that up to 70% of VLM benchmarks are "blindly solvable", i.e. without the image modality, and that converting multiple-choice to generative formats reveals capability drops of up to 35%. This suggest that evaluation/contamination artifacts inflating reported performance *may* be endemic across foundation model domains.

**Disclaimer**

This work aims to initiate a discussion by presenting a critical re-examination of evaluation practices in TLM research, using Tabula-8B as a case study. Our critique targets systemic evaluation patterns across the TLM literature, not the Tabula-8B authors or their work specifically. Indeed, we **selected Gardner et al. (2024) precisely because of the authors' significant commitment to open science**; releasing the T4 training corpus, model weights, and evaluation suite enabled the kind of independent verification we conduct here. A literature review (Appendix G) confirms that the same methodological gaps recur across other major TLM papers. Our goal is constructive, namely to surface and address these issues openly for the benefit of researchers developing TLMs.

## 3. Approach

We operationalize the central question of whether TLMs generalize as three falsifiable sub-questions. First, does TLM performance exceed naive baselines across task types? Second, can top performance be attributed to training data contamination? Third, is tabular pretraining necessary, or does general instruction-following suffice? We address each

in turn through a majority-class baseline, targeted contamination probes of top-performing datasets, and instruction-tuned controls without tabular exposure. We employ Tabula-8B (Gardner et al., 2024) as a representative TLM, re-evaluating it against baselines specifically designed to disentangle distinct sources of performance. The majority-class baseline predicts the most frequent class in each test set, establishing the minimum accuracy threshold and enabling us to compute *lift* (improvement over naive prediction). This is our primary metric (see Section 4.1 for rational). To disentangle instruction-following capability from tabular knowledge, we fine-tuned Llama-3-8B on the 50K Alpaca dataset (Taori et al., 2023), a general-purpose instruction-following corpus containing no tabular data (see Appendix A for training examples and evaluation protocol), and refer to this model as Alpaca. We also evaluate the base Llama-3-8B model without fine-tuning to establish a lower bound and isolate the contribution of instruction-tuning. Crucially, Tabula-8B and Alpaca are parallel fine-tunings from the same Base Llama-3-8B. Tabula-8B is trained on T4 tabular data without general instruction-tuning, while Alpaca is trained on general instructions without any tabular exposure. Any performance gap between the two branches therefore isolates the contribution of tabular pretraining from that of instruction-tuning. For quartile tasks, we additionally evaluate Alpaca+Q, which augments Alpaca with 10K quartile-format examples from 20 randomly held-out quartile datasets to isolate format familiarity from tabular reasoning. Training details and hyperparameters are provided in Appendix A. We use parameter-efficient LoRA fine-tuning (Hu et al., 2022) for compute reasons; prior work suggests LoRA matches full fine-tuning on instruction-tuning at this scale (Dettmers et al., 2023), and any residual gap would, if anything, increase Alpaca's recovery of Tabula-8B's accuracy and strengthen our central finding.

We focus on the UniPredict subset of the Tabula-8B benchmark (Wang et al., 2023), which contains datasets sourced from Kaggle with semantically meaningful column names (e.g., `Workout_Day`) rather than generic identifiers. After excluding 4 datasets for which we could not run inference, our evaluation comprises 165 datasets spanning three task types: binary classification (64 datasets), categorical classification (55 datasets), and quartile classification (46 datasets). See Section 4.1.2 for more details on quartile classification. For Alpaca+Q evaluation, 20 quartile datasets were held out for training, leaving 26 quartile datasets for evaluation. Representative prompt examples for each task type are provided in Appendix A.2.

To investigate data contamination, we searched the T4 training corpus for evaluation examples from top-performing datasets (Table 2), where contamination would be most consequential for performance claims. Specifically, we searched for three patterns: distinctive identifiers such as

names and IDs that uniquely identify records, row-level value combinations with column name variations, and task-level associations (e.g., date-day mappings) which enable memorization-based solutions. Extended methods are provided in Appendix B.

*Table 1.* Performance metrics for "imbalance rider" including raw accuracy (Acc.), majority-class baseline accuracy (Maj.), lift over majority-class baseline (Lift) and Cohen's Kappa ($\kappa$). Aggregated averages across all benchmark datasets presented for comparison.

| DATASET | ACC. | MAJ. | LIFT | $\kappa$ |
|---|---|---|---|---|
| BRAIN STROKE | 0.958 | 0.959 | -0.001 | -0.002 |
| STROKE PREDICTION | 0.949 | 0.949 | +0.000 | 0.000 |
| ALL SPACE MISSIONS | 0.904 | 0.903 | +0.001 | 0.242 |
| BANK LOANS | 0.877 | 0.919 | -0.042 | 0.105 |
| UBER DATA ANALYSIS | 0.845 | 0.932 | -0.087 | 0.167 |
| BANK PERSONAL LOAN | 0.843 | 0.899 | -0.057 | 0.062 |
| AVG. ALL (N=165) | 0.634 | 0.487 | +0.146 | 0.336 |

## 4. Findings

Our findings address three diagnostics, namely missing baselines and task-type composition (Section 4.1), pervasive data contamination (Section 4.2), and the contribution of instruction-following relative to tabular pretraining (Section 4.3).

### 4.1. Missing Baselines and Task-Type Bias Inflate Reported Performance

A fundamental requirement for evaluating any classifier is comparison against a naive baseline (Japkowicz & Shah, 2011). For classification tasks, the majority-class baseline establishes a lower bound a model must exceed to demonstrate learning beyond class-frequency exploitation. This practice is standard in machine learning evaluation (Fernández et al., 2018; He & Garcia, 2009), yet it is notably absent from TLM evaluations. To address this, we begin by computing the majority-class baseline accuracy for all 165 datasets in the Tabula-8B benchmark and report *lift over baseline* rather than raw accuracy.

#### 4.1.1. HIGH ACCURACY DOES NOT IMPLY LEARNING.

The absence of baseline comparison allows datasets with severe class imbalance to masquerade as successes. We term these "imbalance riders", datasets where class imbalance inflates raw accuracy while lift over baseline remains negligible or negative. Table 1 presents 6 examples of these imbalance rider datasets. Tabula-8B achieves high raw accuracy but negligible or negative lift, implying the model learned little beyond predicting the majority class. For instance, on the stroke prediction dataset, Tabula-8B achieves 94.9% accuracy, which appears impressive until one observes that the majority class baseline accuracy is

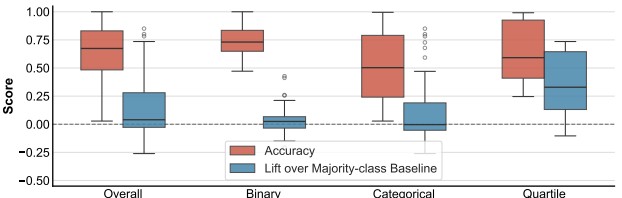

*Figure 1.* Performance decomposition by task type compared to aggregated metrics. Raw accuracy in red and lift over majority-class baseline in blue. Dotted line indicates 0 performance level for respective metrics.

also 94.9%. Alternatively, Uber and Bank appear favorable relative to the cross-dataset mean, yet Tabula-8B underperforms their majority-class baselines by 8.7% and 5.7%, respectively. In total, we see 65/165 (39.4%) datasets where the model yields no positive lift compared to majority class. As further validation, we computed Cohen's Kappa (Cohen, 1960) across all the datasets in Figure 5. We found 12 datasets exhibit negative $\kappa$, indicating agreement worse than chance (Table 6).

#### 4.1.2. TASK-TYPE DECOMPOSITION REVEALS SYSTEMATIC HETEROGENEITY

The aggregate reported benchmark metrics are comprised of three task types: binary classification, categorical (multiclass) classification, and what is labeled quartile "regression." The latter is something of a misnomer. Because language models generate discrete tokens rather than continuous values, TLM evaluations convert regression tasks into classification by discretizing continuous targets into quartile bins and asking the model to predict which bin contains the true value (Gardner et al., 2024; Sun et al., 2024). Thus, all three task types are *classification* problems, and we henceforth refer to quartile "regression" as quartile classification

Figure 1 presents performance stratified by task type. These (and Table 7) results reveal a striking heterogeneity; while quartile classification achieves a median lift of 32.6 percent-

*Table 2.* Top-10 performing datasets for Tabula-8B on the Unipredict benchmark, ranked by accuracy. Metrics include task-type, raw accuracy (Acc.), and lift over the majority baseline (Lift).

| DATASET | TASK | ACC. | LIFT |
|---|---|---|---|
| BUNDESLIGA-SEASONS | BINARY | 100.0% | 41.2% |
| US-WOMENS-LABOR | BINARY | 99.6% | 42.8% |
| UDEMY-COURSES | CATEG. | 99.5% | 67.9% |
| PELOTON-DATA | CATEG. | 99.4% | 78.2% |
| BITCOIN-PRICE | QUANT. | 99.1% | 72.9% |
| NETFLIX-STOCK | QUANT. | 99.0% | 73.5% |
| BITCOIN-USD | QUANT. | 99.0% | 70.7% |
| POKEMON | CATEG. | 99.0% | 85.0% |
| YAHOO-STOCK | QUANT. | 98.8% | 73.1% |
| TESLA-STOCK | QUANT. | 98.8% | 72.1% |

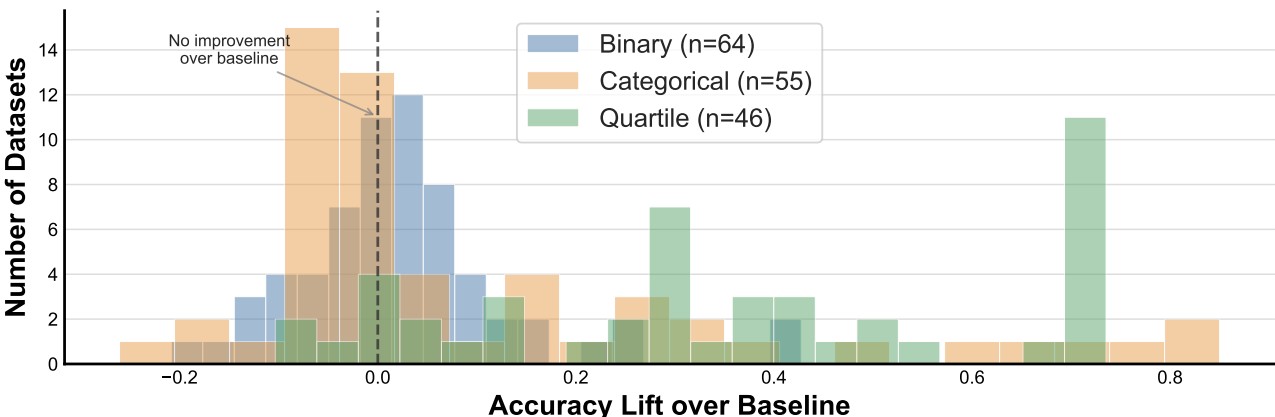

*Figure 2.* Distribution of accuracy lift over majority-class baseline accuracy lift over task type. Dotted line indicate 0 accuracy lift.

age points over baseline, binary classification achieves only 2.5 percentage points, barely exceeding majority classification. More concerning, categorical classification exhibits a *negative* (-0.3) median lift, indicating that TLMs can perform worse than the majority class baseline on more than half of these datasets. Furthermore, the asymmetry in failure rates in Table 7 is equally revealing. Of the 65 datasets where Tabula-8B performs worse than or equal to baseline, 30 are binary classification and 30 are categorical, but only 5 are quartile classification. This pattern suggests that whatever the model has learned transfers poorly to standard classification tasks. Lastly, a one-way ANOVA confirms that performance differences across task types are statistically significant ($F = 30.8$, $p < 10^{-9}$), with all pairwise comparisons also confirming heterogeneity (Table 8).

Figure 2 also illustrates how quartile classification tasks dominate the upper tail of performance distribution. Notably, 11 of the top-20 performing datasets are stock price prediction tasks, all formatted as quartile classification. We find direct evidence in Sections 4.2 and 4.3 that structural properties of these tasks, such as leakage from correlated features and exploitable format/structure, may make them artificially easy independent of tabular reasoning (Appendix D).

**Summary.** Stratified evaluation reveals that TLM performance on binary and categorical classification is marginal, with 39.4% of datasets showing zero or negative lift over majority-class baseline. Strong aggregate accuracy is driven by quartile classification, a task format with structural properties that may inflate performance independent of tabular reasoning (Appendix D). Similar baseline omissions and lack of task-type stratification pervade TLM evaluations broadly. Our literature review of other TLM papers found none report majority-class baselines or chance-corrected metrics (Appendix G), warranting wider re-examination.

### 4.2. Data Leakage is Pervasive

TLM evaluations typically acknowledge some degree of overlap but assume minimal impact on performance (Section 2). However, this assumption has not been rigorously tested for TLMs. We investigated this claim by searching the T4 (Gardner et al., 2024) training corpus for evaluation examples from top-performing datasets (Table 2), where contamination would be most consequential for performance claims (Appendix B). Our analysis reveals three distinct forms of leakage, each with different implications for detectability and mitigation.

#### 4.2.1. COMPLETE TRAIN-TEST OVERLAP

The most direct form of contamination occurs when evaluation examples appear verbatim in training data. We find this pattern in the `us-womens-labor-force-participation` dataset, which contains 753 observations with domain-specific features such as `kids5` (children under 5), `hushrs` (husband's hours worked), and `lfp` (labor force participation, the prediction target).

Searching T4 for these distinctive column names revealed *all 753 evaluation rows* in the training corpus. Figure 3(a) shows a representative example: test row 728 matches a training record with identical values across all fields. While column names differ slightly between sources (e.g., `k618` vs `kids618`), the data and labels are identical. This means the model has seen every evaluation example, including its label, during training. The 99.6% accuracy on this dataset thus cannot be interpreted as evidence of generalization. Additional matches are provided in Table 9 (Appendix E).

#### 4.2.2. DIRECT CONTAMINATION WITH LABEL EXPOSURE

A second form of leakage occurs when evaluation records appear across multiple T4 tables, often with the *target* label

explicitly included. This pattern is pervasive in the stock price datasets that dominate the top performers.

**Financial time series.** We examined the Bitcoin price prediction task, which asks the model to predict the `Close` price (discretized into quartile bins) given features including `Date`, `Open`, `High`, `Low`, and `Adj Close`. Searching T4 for test examples revealed extensive duplication: individual records appear in up to four separate T4 chunks, each containing the target `Close` value. Figure 6 (Appendix E) shows two representative test examples and their T4 matches. The first example (2019-05-27) appears in three T4 data chunks; the second (2021-04-20) appears in four. In every case, the training data includes the exact `Close` value the model is asked to predict. This pattern extends across the Netflix, Bitcoin-USD, Yahoo and Tesla stock datasets that collectively account for 5 of the top 10 benchmark performers (Figure 7 in Appendix E). These tasks also suffer from a compounding problem: even without contamination, the task format enables trivial shortcuts. When `Adj Close ≈ Close` (as is typical for recent records without stock splits or dividends), the model can infer the answer from the input features alone. Near-perfect accuracy on these tasks thus reflects some combination of memorization and numeric leakage rather than learned price forecasting.

**Pokemon type classification.** The `pokemon` dataset tasks the model with predicting a Pokémon's primary type (`Type 1`) from its name and battle statistics. Tabula-8B achieves 99.0% accuracy, the highest lift (85%) over the majority class baseline among all datasets evaluated. To investigate potential data contamination, we searched for test examples in the T4 pretraining corpus. To reduce false positives from common Pokémon names appearing in non-tabular contexts, we focused on the 98 examples that Alpaca (see Section 4.3) misclassified. All 98 were found in T4 with their correct labels, suggesting Tabula's advantage on these examples may stem from memorization during pretraining.

### 4.2.3. TASK LEAKAGE BEYOND ROW-LEVEL DEDUPLICATION

The most subtle form of contamination occurs when the evaluation dataset itself is absent from training, yet the *task* remains trivially solvable from training data. Standard deduplication methods which match evaluation rows against

*Table 3.* Summary of contamination (types) across top-performing Tabula-8B datasets, by raw accuracy (Acc.), in Unipredict.

| DATASET | TASK | ACC. | TYPE |
|---|---|---|---|
| US-WOMENS-LABOR | BINARY | 99.6% | COMPLETE |
| PELOTON-DATA | CATEG. | 99.4% | TASK LEAKAGE |
| POKEMON | CATEG. | 99.0% | COMPLETE |
| FORCASTING(×5) | QUART. | ≥98.8% | DIRECT+LABELS |

**(a) Complete Overlap: `us-womens-labor`**

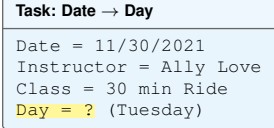

```
Test Row 728

Unnamed:  0 = 728
kids618 = 3, age = 41
nwifeinc = 77.0
lfp = ?
```

```
T4 Match

Unnamed:  0 = 728
k618 = 3, age = 41
inc = 77.0
lfp = no  (Label exposed)
```

753/753 rows in T4 with labels exposed.

**(b) Task Leakage: `peloton-data`**

```
Task: Date → Day

Date = 11/30/2021
Instructor = Ally Love
Class = 30 min Ride
Day = ?  (Tuesday)
```

```
T4: toronto-transit-delays

Date:  2021-11-30
Day:  Tuesday
Line:  501, ...
(844 matches in unrelated tables)
```

No Peloton records in T4; task solvable via memorization.

*Figure 3.* Examples of data contamination in the T4 and Unipredict datasets. (a) Complete Overlap: A test row from `us-womens-labor` perfectly matches a T4 record, exposing the label. (b) Task Leakage: In `peloton-data`, while passing row-level deduplication, the date-to-day mapping is encoded in 844 unrelated T4 records; a representative match is shown, enabling solvability via memorization.

training rows can fail to detect this form of leakage.

The `peloton-data` dataset illustrates this phenomenon. The task requires predicting the day of the week (`Workout_Day`) from features including `Workout_Date` (e.g., "11/30/2021") and workout metadata. Searching T4 for specific Peloton records using `Workout_ID`, timestamps, and instructor names yielded *no exact* matches. Hence, by standard criteria, this dataset is "clean."

However, searching T4 for a sample date-day pair from the test set revealed that there is indeed contamination. Querying for records containing both "2021-11-30" and "Tuesday" returned 844 matches from *unrelated* tables. These tables have timestamped records that happen to include both the date and its corresponding day of week (Figure 3(b); extended examples in Table 10 in Appendix E). This constitutes *task leakage*: the model can recall memorized date-day associations from hundreds of unrelated T4 tables rather than performing calendar arithmetic. The prediction task may thus reduce to recalling memorized associations rather than performing calendar arithmetic. This form of contamination is particularly concerning because it (a) evades standard deduplication, requiring task-specific analysis to detect, and (b) can inflate performance on any task reducible to widely-attested factual associations.

**Summary** Table 3 consolidates our findings. In aggregate, 8 of the top-10 datasets we examined exhibit some form of contamination, and 11 of the top-20 are stock-format quartile tasks of the type where we documented direct label exposure. The evidence reveals a consistent pattern that the highest-accuracy datasets in the Tabula-8B benchmark exhibit substantial contamination. These results have two implications. First, strong performance on these datasets

cannot be interpreted as evidence of tabular reasoning. Second, the diversity of contamination mechanisms[4] suggests that standard row-level deduplication is insufficient for tabular benchmarks. Our search is targeted at top performers and therefore demonstrates the presence, not the absence, of contamination. Combined with Section 4.3, where Tabula's advantage disappears on non-contaminated quartile datasets (Table 5), the observed pattern is consistent with memorization rather than coincidental overlap.

### 4.3. Instruction-Following, Not Tabular Knowledge, Drives Performance

The Tabula-8B authors attribute their model's strong performance to fine-tuning on the T4 corpus, implying that exposure to millions of serialized tables teaches tabular reasoning. We test an alternative hypothesis: performance gains primarily reflect learning to follow instructions in a particular *format*, rather than acquiring tabular knowledge. To isolate these factors, we evaluate Tabula-8B against Base Llama-3-8B, Alpaca, and, for quartile tasks, Alpaca+Q,(see Section 3 and Appendix A). Because Tabula-8B and Alpaca are parallel fine-tunings from the same Base Llama (Section 3), this structure decomposes Tabula-8B's performance into contributions from instruction-tuning versus tabular pretraining. All models use greedy decoding and 4-shot prompting with identical serialized rows; only the prompt template differs.

#### 4.3.1. CLASSIFICATION: INSTRUCTION-TUNING DOMINATES

Figure 4 and Table 4 present our main results across 119 classification datasets[5]. Mean accuracy progresses from 47.7% (Base Llama) to 58.6% (Alpaca) to 63.5% (Tabula-8B), a total improvement of 15.9 percentage points from Base Llama to Tabula-8B. Critically, instruction-tuning alone accounts almost 11 points of this gain (69%), while tabular pretraining contributes only 4.9 points (31%). Thus, two-thirds of Tabula-8B's improvement over the base model is attributable to general instruction-following capability, not tabular-specific knowledge.

The Base Llama model performs worse than majority-class prediction on 75.6% of datasets, with a mean lift of −9.4 points. This establishes that the pretrained model, despite its general capabilities, cannot effectively perform tabular classification without some instruction(/fine)-tuning. Alpaca, fine-tuned only on general instructions with no tabular exposure, dramatically improves performance, exceeding the majority baseline on 39.5% of datasets (compared to Base Llama's 24.4%) and achieves 92.2% of Tabula-8B's mean accuracy. Alpaca outperforms Base Llama on 79.8% of clas-

---

[4]See Limitations in Appendix B

[5]Values in this section rounded to one decimal place.

*Table 4.* Aggregate performance across binary and categorical tasks for Tabula-8B, Base Llama, and Alpaca. Metrics include accuracy (Acc.), lift over majority baseline (Lift), the percentage of Tabula-8B accuracy recovered by alternative models (Recovery), and percentage of datasets performing at or below the majority-class baseline ($\leq$ Baseline).

| MODEL | ACC. | LIFT | RECOVERY | $\leq$ BASELINE |
|---|---|---|---|---|
| BASE LLAMA | 47.7 | −9.4 | 75.0 | 75.6 |
| ALPACA | 58.6 | +1.6 | 92.2 | 60.5 |
| TABULA-8B | 63.5 | +6.5 | NA | 50.4 |

sification datasets, confirming that instruction-tuning, not tabular pretraining, accounts for a large portion of the performance recovery. While the precise mechanism remains unclear, we hypothesize that instruction-tuning equips the model with general capabilities for comprehending task descriptions and following input-output mappings, skills that may prove sufficient for many tabular classification tasks without requiring explicit tabular exposure. Additionally, despite having never seen tabular data, Alpaca achieves competitive lift to Tabula-8B with median lift of −0.7 on binary tasks versus Tabula's 2.5, and −1.8 on categorical versus Tabula's −0.3. Both distributions cluster near zero, with substantial mass at or below the baseline. On 38.7% of datasets, Alpaca matches or exceeds Tabula-8B's accuracy.

The positive mean lift for Tabula-8B (6.5 v Alpaca's 1.6) is driven by a number of outliers. Eight of the top-10 datasets contributing to this gap exhibit the contamination patterns documented in Section 4.2. In the remaining 2, Alpaca nearly matches Tabula's performance (Table 11). Excluding 5 contaminated classification datasets, the instruction-tuning contribution rises to 73%, with tabular training accounting for only 27% of improvement (Table 12).

#### 4.3.2. QUARTILE CLASSIFICATION: FORMAT AND CONTAMINATION EXPLAIN THE GAP

Quartile tasks initially appear to show a substantial Tabula advantage with mean lift of 34.0% versus Alpaca's 5.6%(Figure 4). Base Llama performs poorly on these tasks, with mean lift of −16.7 points, confirming that the quartile bin format is entirely unfamiliar to the pretrained model. However, we can decompose this 28.5 point gap between Tabula and Alpaca into two components: format unfamiliarity and data contamination.

First, Alpaca+Q, which adds only quartile-format examples without any tabular pretraining, achieves mean lift of 25.8%, closing 71.3% of the gap to Tabula. This demonstrates that the apparent advantage primarily reflects familiarity with the quartile bin output format, not learned tabular reasoning. Second, the remaining 8.2 point gap is not uniformly distributed. Decomposing by contamination risk (Table 5) reveals that Tabula retains a 17.9 point advantage on stock/-

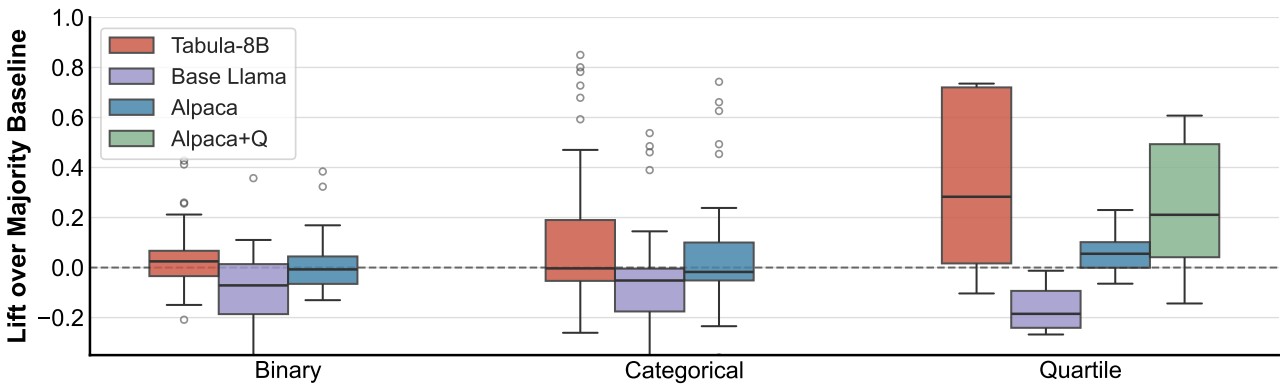

*Figure 4.* Lift over majority-class baseline by task type for Tabula-8B, Base Llama, Alpaca, and Alpaca+Q. Alpaca+Q was instruction-tuned for quartile classification only is not applicable to other tasks. The dotted line indicates the zero performance level relative to the majority-class baseline.

financial datasets, which are widely replicated across web corpora and where we documented direct contamination in Section 4.2. In stark contrast, on non-stock datasets, the gap reverses this with Alpaca+Q achieves 6.7pp lift versus Tabula's 6.5pp, a difference of −0.1pp. When contamination risk is controlled, instruction-tuning with format exposure fully recovers Tabula-8B's performance.

**Summary.** Instruction-tuning on general-purpose data, without any tabular exposure, can account for about 69% of Tabula-8B's improvement over the base model on classification tasks. Format familiarity closes the remaining gap on quartile classification. The residual advantage exists only on stock datasets with documented contamination. Taken together, these findings suggest that format adaptation and memorization, rather than learned tabular reasoning, explains the strong performance attributed to tabular pretraining.

## 5. Recommendations for TLM Evaluation

The preceding findings reveal systematic issues in TLM evaluations and interpretation. Our literature review (Appendix G) found that these patterns extend well beyond just Tabula-8B. So, we now distill these into actionable recommendations the TLM community which would help

*Table 5.* Aggregate performance on quartile tasks partitioned by stock/financial and non-stock categories. Columns report dataset counts ($N$), lift over majority baseline for Tabula-8B and Alpaca+Q models, and the resulting performance difference (Gap) in percentage points.

| DATASETS | N | TABULA | ALPACA+Q | GAP |
|---|---|---|---|---|
| STOCK/FINANCIAL | 12 | 66.1 | 48.2 | 17.9 |
| NON-STOCK | 14 | 6.5 | 6.7 | −0.1 |
| ALL QUARTILE | 26 | 34.0 | 25.8 | 8.2 |

distinguish genuine advances from false signal.

**R1: Report baseline comparisons and broader metrics.** *(motivated by Section 4.1, Table 1, and Figure 5).* Raw accuracy is uninterpretable without reference to task difficulty. We recommend reporting performance relative to (1) a majority-class baseline, (2) a base model without the proposed contributions, and (3) Cohen's Kappa (Cohen, 1960) or similar metrics robust to class imbalance (Japkowicz & Shah, 2011).

**R2: Stratify results by task type.** *(motivated by Section 4.1.2, Figure 1, and Tables 7 and 8).* Aggregating performance across heterogeneous task types (binary, multiclass, regression-as-classification) obscures systematic strengths and weaknesses. We recommend that benchmark papers report performance separately for each task type, with statistical tests for heterogeneity (e.g., ANOVA (Fisher & Mackenzie, 1923; Girden, 1992)). This practice is standard in meta-analysis (Higgins et al., 2003) and enables readers to assess where models succeed and fail.

**R3: Release evaluation code for public verification of task construction.** *(motivated by Section 4.1.2 and Appendix D).* When continuous targets are discretized into classification tasks, the resulting problem may be trivially solvable via shortcuts that do not reflect genuine tabular reasoning. To enable independent verification, we recommend that authors release (1) evaluation code that computes all reported metrics, (2) raw predictions on benchmark datasets, and (3) baseline implementations. This transparency allows the community to audit for numeric leakage, evaluation artifacts, and other confounds, facilitating the kind of reanalysis we conducted here.

**R4: Conduct contamination analysis beyond row-level deduplication.** *(motivated by Section 4.2, Table 3, and Figure 3).* Standard row-matching is insufficient for tabular data, which can appear under varying column names, across multiple tables, or as entity-level associations. We recom-

mend reporting decontamination methodology, proportion of training corpus searched, and results of entity-level and fuzzy matching. Datasets from widely-replicated sources (stock prices, Kaggle competitions) should be assumed contaminated absent evidence otherwise, and should generally be excluded from benchmarks.

**R5: Audit whether tasks require tabular reasoning.** *(motivated by Section 4.2.3 and the* `udemy-courses` *footnote on Table 12).* High accuracy does not imply tabular reasoning if tasks reduce to factual recall, text classification, or logical inference from input features. Authors should search training corpora for underlying associations (not just evaluation rows), verify tasks cannot be solved through surface-level text understanding, and exclude tasks that do not require tabular-specific (or multi-feature) reasoning.

**R6: Compare against instruction-tuned baselines without tabular exposure.** *(motivated by Section 4.3, Table 4, and Figure 4).* Comparing TLMs against base models only conflates instruction-following capability with tabular knowledge. We recommend including at least one instruction-tuned baseline (such as Alpaca (Taori et al., 2023), FLAN (Wei et al., 2022)) with no tabular exposure. If a TLM does not substantially outperform this baseline, claims of tabular reasoning require additional justification.

**R7: Foster critical evaluation as reviewers and practitioners.** *(synthesized from all of Section 4).* Responsibility for rigorous evaluation extends beyond authors to the broader community that reviews, builds on, and interprets TLM research. Reviewers should expect baseline comparisons, task-stratified results, and contamination analysis as standard practice. Near-perfect accuracy on common datasets or tasks with severe class imbalance should prompt scrutiny. When a task reduces to text classification or feature inference, strong performance is not evidence of tabular reasoning. Reviewers should also ask whether claimed capabilities match task/evaluation requirements.

## 6. Limitations

Our evaluation framework does warrant discussion of several limitations. First, our contamination analysis searched for specific patterns, such as distinctive column names, entity identifiers (e.g., Pokemon names), and value combinations. An exhaustive search of T4's approximately 150 zip archives containing millions of parquet files was not tractable for us[6]. Consequently, our analysis can only demonstrate the *presence* of contamination, not its absence. Contamination detection for tabular data lacks established best practices and row-level deduplication is insufficient (as we demonstrated), but comprehensive alternatives remain

---

[6]Agentic frameworks, as used in other domains (Youngblut et al., 2025), can make this possible in the future

an open problem. Machine unlearning could in principle provide a causal test of contamination's effect on accuracy, though current methods are not yet reliable at the 100B-token corpus scale of T4.

Next, our analysis focused on the UniPredict subset (165 datasets after exclusions) rather than the full Tabula-8B benchmark (329 datasets). Additionally, we evaluated only classification tasks. Other TLM capabilities such as imputation, generation, and multi-table reasoning were not assessed. We believe the current scope is sufficient as a *proof-of-principle* but more expansive re-evaluation of other benchmarks should be the focus of future work.

Lastly, this investigation focused on Tabula-8B and the T4 corpus as a feasible case study. Other TLMs trained on different corpora or with different objectives may exhibit different patterns (strengths and failure modes). That said, our survey of the literature suggests that metric aggregation, baseline omission and lack of sufficient contamination analysis are endemic across TLM evaluations, implying these concerns very likely extend beyond the specific context we examined.

## 7. Conclusions

Our systematic investigation reveals that the strong performance attributed to TLMs likely does not reflect genuine tabular generalization. These results are established directly for Tabula-8B on the UniPredict benchmark. We extend them to the broader TLM literature as a structural conjecture, grounded in our scorecard of five major TLM papers (Appendix G), which shows the same evaluation gaps recurring across the field. Specifically, we find that aggregate metrics are inflated by task-type composition (with quartile classification driving gains), that top-performing datasets exhibit pervasive data contamination, and that instruction-tuning alone, without any tabular exposure, recovers the majority of reported performance, with residual gaps attributable to data contamination rather than learned tabular reasoning. These findings are not intended to discourage research on tabular foundation models, but rather to help build more trustworthy TLMs on solid empirical foundations. Our seven recommendations provide concrete steps toward evaluation standards capable of distinguishing genuine advances from artifacts. Ultimately, we hope this analysis contributes to the field's progress, offering empirically grounded recommendations that foster more robust standards to aid both the researchers advancing these architectures and the practitioners deciding on their deployment.

## Impact Statement

This paper presents work whose goal is to advance the field of machine learning through more rigorous evaluation practices. By identifying systematic evaluation issues in Tabular Language Model research, we aim to prevent overestimation of model capabilities that could lead to inappropriate deployment in high-stakes domains. Our critique targets systemic patterns across the literature rather than individual work, and is intended to strengthen the empirical foundations of the field for the benefit of both researchers and practitioners.

## Acknowledgments

We thank IT University of Copenhagen's HPC and P1 DTU HPC for computational resources. We also thank Sriram Sankararaman and Jonathan Flint for their helpful comments and suggestions on an early draft of this manuscript.

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

# A. Alpaca Details

## A.1. Alpaca Training

To disentangle instruction-following capability from tabular knowledge, we fine-tuned the base Llama-3-8B model on 50K examples from the Alpaca dataset (Taori et al., 2023)[7], a general-purpose instruction-following corpus containing no tabular data. We used parameter-efficient LoRA fine-tuning (Hu et al., 2022) with rank $r = 8$ and scaling factor $\alpha = 16$, applying LoRA adapters to attention projections (q_proj, v_proj, output_proj) and MLP layers. Training proceeded for a single epoch with a learning rate of $2 \times 10^{-4}$ and batch size of 128.

At inference time, we reformatted the Tabula-8B evaluation examples from their custom special-token format into the standard Alpaca instruction template. Specifically, we replaced Tabula's <|endinput|> and <|endcompletion|> delimiters with the Alpaca format:

```
### Instruction:
{task description}

### Input:
{serialized table row}

### Response:
```

The serialized row content and few-shot examples were preserved exactly as in the Tabula evaluation, ensuring that any performance differences reflect model capabilities rather than input formatting.

For reference, we show two representative training examples from the Alpaca dataset. These examples illustrate the general instruction-following format and contain no tabular data:

```
### Instruction:
Give three tips for staying healthy.

### Input:

### Response:
1. Eat a balanced and nutritious diet: Make sure your meals are
inclusive of a variety of fruits and vegetables, lean protein,
whole grains, and healthy fats. This helps to provide your body
with the essential nutrients to function at its best and can help
prevent chronic diseases.

2. Engage in regular physical activity: Exercise is crucial for
maintaining strong bones, muscles, and cardiovascular health. Aim
for at least 150 minutes of moderate aerobic exercise or 75
minutes of vigorous exercise each week.

3. Get enough sleep: Getting enough quality sleep is crucial for
physical and mental well-being. It helps to regulate mood, improve
cognitive function, and supports healthy growth and immune
function. Aim for 7-9 hours of sleep each night.

                                ...

### Instruction:
Explain why the following fraction is equivalent to 1/4

### Input:
4/16

### Response:
The fraction 4/16 is equivalent to 1/4 because both fractions
```

---

[7]https://huggingface.co/datasets/yahma/alpaca-cleaned

represent the same value. A fraction can be simplified by dividing
both the numerator and the denominator by a common factor. In this
case, 4 is a common factor of both the numerator and the
denominator of 4/16. When we divide both by 4, we get 4/4 = 1 and
16/4 = 4, so the simplified fraction is 1/4.

## A.2. Prompt Examples

We provide representative prompt examples for each task type to illustrate the input format used in our evaluation. All
examples use 4-shot prompting, where four labeled examples precede the test instance. We show both the Tabula-8B format
(using special tokens) and our Alpaca format (using instruction template delimiters).

### A.2.1. CATEGORICAL CLASSIFICATION: POKÉMON TYPE PREDICTION

The pokemon dataset predicts a Pokémon's primary type (Type 1) from its name and battle statistics.

**Tabula-8B Format:**

```
<|begin_of_text|>Predict the value of Type 1: ||Steel||Normal||
Rock||...||Flying|| The # is 718. The Name is Zygarde50% Forme.
The Type 2 is Ground. The Total is 600. The HP is 108. The Attack
is 100. The Defense is 121. The Sp Atk is 81. The Sp Def is 95.
The Speed is 95. The Generation is 6. The Legendary is True.
What is the value of Type 1? ||Steel||Normal||...||Flying||
<|endinput|>Dragon<|endcompletion|>
[3 more few-shot examples]
Predict the value of Type 1: ... The # is 319. The Name is
Sharpedo. ...<|endinput|>
```

**Alpaca Format:**

```
Below is an instruction that describes a task. Write a response
that appropriately completes the request.

### Instruction:
Predict the correct value based on the input.

### Input:

Example 1:
Predict the value of Type 1: ||Steel||Normal||Rock||Grass||Bug||
Ground||Fairy||Water||Electric||Ghost||Dragon||Ice||Dark||Fire||
Fighting||Psychic||Poison||Flying|| The # is 718. The Name is
Zygarde50% Forme. The Type 2 is Ground. The Total is 600. The HP
is 108. The Attack is 100. The Defense is 121. The Sp Atk is 81.
The Sp Def is 95. The Speed is 95. The Generation is 6. The
Legendary is True. What is the value of Type 1? ||Steel||Normal||
Rock||Grass||Bug||Ground||Fairy||Water||Electric||Ghost||Dragon||
Ice||Dark||Fire||Fighting||Psychic||Poison||Flying||
Response: Dragon

[Examples 2-4 follow same format]

Now complete the following:
Predict the value of Type 1: ||Steel||Normal||Rock||...||Flying||
The # is 319. The Name is Sharpedo. The Type 2 is Dark. The Total
is 460. The HP is 70. The Attack is 120. The Defense is 40. The
Sp Atk is 95. The Sp Def is 40. The Speed is 95. The Generation
is 3. The Legendary is False. What is the value of Type 1?
||Steel||Normal||Rock||...||Flying||

### Response:
```

## A.2.2. BINARY CLASSIFICATION: US WOMEN'S LABOR FORCE PARTICIPATION

The `us-womens-labor-force-participation` dataset predicts labor force participation (`lfp`) from demographic features.

**Tabula-8B Format:**

```
<|begin_of_text|>Predict the value of lfp: ||1||0|| The Unnamed:
0 is 235. The hours is 1640. The kids5 is 0. The kids618 is 1.
The age is 38. The educ is 12. The wage is 3.658. The repwage is
3.25. The hushrs is 2040. The husage is 46. The huseduc is 12.
The huswage is 3.186. The faminc is 12500. The mtr is 0.721. The
motheduc is 7. The fatheduc is 7. The unem is 7.5. The city is 0.
The exper is 19. The nwifeinc is 6.5. The wifecoll is FALSE. The
huscoll is FALSE. What is the value of lfp? ||1||0||<|endinput|>1
<|endcompletion|>...[3 more examples]...<|endinput|>
```

**Alpaca Format:**

```
Below is an instruction that describes a task. Write a response
that appropriately completes the request.

### Instruction:
Predict the correct value based on the input.

### Input:

Example 1:
Predict the value of lfp: ||1||0|| The Unnamed: 0 is 235. The
hours is 1640. The kids5 is 0. The kids618 is 1. The age is 38.
The educ is 12. The wage is 3.658. The repwage is 3.25. The
hushrs is 2040. The husage is 46. The huseduc is 12. The huswage
is 3.186. The faminc is 12500. The mtr is 0.721. The motheduc is
7. The fatheduc is 7. The unem is 7.5. The city is 0. The exper
is 19. The nwifeinc is 6.5. The wifecoll is FALSE. The huscoll is
FALSE. What is the value of lfp? ||1||0||
Response: 1

[Examples 2-4 omitted for brevity]

Now complete the following:
Predict the value of lfp: ||1||0|| The Unnamed: 0 is 728. The
hours is 0. The kids5 is 0. The kids618 is 3. The age is 41. The
educ is 12. The wage is 0.0. The repwage is 0.0. The hushrs is
2450. The husage is 48. The huseduc is 15. The huswage is 26.53.
The faminc is 77000. The mtr is 0.442. The motheduc is 7. The
fatheduc is 14. The unem is 14.0. The city is 1. The exper is 8.
The nwifeinc is 77.0. The wifecoll is FALSE. The huscoll is TRUE.
What is the value of lfp? ||1||0||

### Response:
```

## A.2.3. QUARTILE CLASSIFICATION: BITCOIN PRICE PREDICTION

The `bitcoin-price-2014-2023` dataset predicts the closing price discretized into quartile bins, illustrating the quartile classification format.

**Tabula-8B Format:**

```
<|begin_of_text|>Predict the value of Close: ||between 7697.924072
and 20297.0288085||less than 764.11325075||between 764.11325075
and 7697.924072||greater than 20297.0288085|| The Date is
2023-07-09. The Open is 30291.611328125. The High is
30427.58984375. The Low is 30085.591796875. The Adj Close is
```

```
30171.234375. The Volume is 7903327744. What is the value of
Close? ||between 7697.924072 and 20297.0288085||less than
764.11325075||between 764.11325075 and 7697.924072||greater than
20297.0288085||<|endinput|>greater than 20297.0288085
<|endcompletion|>...[3 more examples]...<|endinput|>
```

**Alpaca Format:**

```
Below is an instruction that describes a task. Write a response
that appropriately completes the request.

### Instruction:
Predict the correct value based on the input.

### Input:

Example 1:
Predict the value of Close: ||between 7697.924072 and
20297.0288085||less than 764.11325075||between 764.11325075 and
7697.924072||greater than 20297.0288085|| The Date is 2023-07-09.
The Open is 30291.611328125. The High is 30427.58984375. The Low
is 30085.591796875. The Adj Close is 30171.234375. The Volume is
7903327744. What is the value of Close? ||between 7697.924072 and
20297.0288085||less than 764.11325075||between 764.11325075 and
7697.924072||greater than 20297.0288085||
Response: greater than 20297.0288085

[Examples 2-4 omitted for brevity]

Now complete the following:
Predict the value of Close: ||between 7697.924072 and
20297.0288085||less than 764.11325075||between 764.11325075 and
7697.924072||greater than 20297.0288085|| The Date is 2019-05-27.
The Open is 8674.072265625. The High is 8907.1748046875. The Low
is 8668.705078125. The Adj Close is 8805.7783203125. The Volume
is 27949840384. What is the value of Close? ||between 7697.924072
and 20297.0288085||less than 764.11325075||between 764.11325075
and 7697.924072||greater than 20297.0288085||

### Response:
```

### A.3. Alpaca+Q Training

For quartile classification tasks, the standard Alpaca model performed poorly due to unfamiliarity with the quartile bin output format. These tasks require predicting labels such as "less than 25.3" or "between 25.3 and 50.1," a format not present in the general Alpaca training data.

To isolate format learning from tabular knowledge, we created Alpaca+Q by augmenting the Alpaca training data with 10K quartile-format examples. These examples were drawn from 20 regression datasets that we held out from all evaluation analyses reported in Section 4.3. For each held-out dataset, we discretized the continuous target into quartile bins and formatted examples using the same serialization as Tabula-8B evaluation. This ensures that any performance gains from Alpaca+Q reflect familiarity with the quartile output format rather than exposure to evaluation data.

The 20 held-out datasets used for Alpaca+Q training are:

1. aakashjoshi123/exercise-and-fitness-metrics-dataset
2. altruistdelhite04/gold-price-data
3. amirhosseinmirzaie/countries-life-expectancy
4. arnabchaki/data-science-salaries-2023
5. arslanr369/roblox-stock-pricing-2021-2023
6. awaiskaggler/insurance-csv
7. dsfelix/us-stores-sales

8. `gyanprakashkushwaha/laptop-price-prediction-cleaned-dataset`
9. `harshitshankhdhar/imdb-dataset-of-top-1000-movies-and-tv-shows`
10. `hemanthhari/psycological-effects-of-covid`
11. `iamsumat/spotify-top-2000s-mega-dataset`
12. `mirichoi0218/insurance`
13. `noordeen/insurance-premium-prediction`
14. `oles04/top-leagues-player`
15. `prevek18/ames-housing-dataset`
16. `shreyapurohit/anime-data`
17. `suraj520/dairy-goods-sales-dataset`
18. `teertha/ushealthinsurancedataset`
19. `warcoder/earthquake-dataset`
20. `whenamancodes/students-performance-in-exams`

### A.4. Evaluation Protocol

For all models (Base Llama, Alpaca, Alpaca+Q, and Tabula-8B), we used the same evaluation protocol: few-shot prompting with serialized table rows, followed by extracting the model's predicted class from the generated completion. We used greedy decoding (temperature = 0) to ensure reproducibility. All comparisons use lift over the majority-class baseline as the primary metric, which controls for class imbalance and enables fair comparison across datasets with varying class distributions.

## B. Contamination Detection Methodology

### B.1. Challenges for Tabular Contamination Detection

Data contamination detection in tabular settings poses unique challenges not present in text or image domains. Standard approaches based on exact string matching or $n$-gram overlap (Brown et al., 2020) are insufficient for several reasons:

1. **Permutation invariance.** Tabular data is row-permutation and column-permutation invariant. The same dataset can appear with columns in different orders or rows shuffled, defeating exact-match detection.
2. **Column name variations.** The same underlying data frequently appears across web sources with different column names. For example, we found the `us-womens-labor` dataset with columns named `kids618` in the evaluation set but `k618` in T4, and `nwifeinc` versus `inc` for the same field.
3. **Multi-source duplication.** Popular datasets (stock prices, well-known Kaggle competitions) are replicated across many websites, often with slight schema variations. A single evaluation record may appear in dozens of T4 tables under different presentations.
4. **Task-level leakage.** Even when specific evaluation records are absent from training, the *task* may be solvable from training data. For instance, predicting day-of-week from a date requires no dataset-specific knowledge if the model has memorized date-day associations from other sources.

### B.2. Search Procedure

We focused our contamination analysis on the top-performing datasets (Table 2), where contamination would be most consequential for claims of strong tabular reasoning. The T4 corpus comprises approximately 150 zip archives containing millions of parquet files. Due to computational constraints, we conducted targeted searches rather than exhaustive corpus-wide matching.

For each dataset examined, we employed the following search strategies:

**Distinctive identifier search.** For datasets with unique identifiers (e.g., Pokemon names, course IDs, stock ticker symbols), we searched T4 for these identifiers to locate potentially matching tables. This approach reduces false positives from common values while efficiently locating relevant training data.

**Row-level value matching.** We searched for combinations of field values that would be unlikely to co-occur by chance. For the `us-womens-labor` dataset, we searched for records containing the distinctive column names (`kids5`, `hushrs`,

`nwifeinc`) and verified field-level matches for identified records.

**Task-level association search.** For tasks reducible to factual associations (e.g., date → day-of-week), we searched T4 for the underlying associations rather than specific evaluation records. For the `peloton-data` dataset, we queried for records containing both a sample date ("2021-11-30") and its corresponding day ("Tuesday"), finding 844 matches from unrelated tables.

### B.3. Limitations

Our contamination analysis has several limitations:

1. **Non-exhaustive search.** We searched for specific patterns rather than conducting corpus-wide exact matching. Our analysis can demonstrate the *presence* of contamination but not definitively establish its absence.
2. **Focus on top performers.** We prioritized datasets where contamination would most affect performance claims. Lower-performing datasets were not systematically examined.
3. **Detection methodology limitations.** Tabular contamination detection lacks established best practices. More sophisticated approaches (e.g., fuzzy matching, semantic similarity) may reveal additional contamination we did not detect.

These limitations suggest our findings represent a lower bound on the true extent of contamination in the benchmark.

## C. Extended Task-heterogeneity Results

*Table 6.* Datasets with negative Cohen's Kappa, indicating classifier agreement worse than chance. Metrics include raw accuracy (Acc.), majority-class baseline accuracy (Maj.), lift over majority-class baseline (Lift) and Cohen's Kappa ($\kappa$).

| DATASET | TYPE | ACC. | MAJ. | LIFT | $\kappa$ |
|---|---|---|---|---|---|
| hawkingcr/airbnb-for-boston-with-fraud-detection | Binary | 0.749 | 0.792 | -0.043 | -0.051 |
| tejashvi14/employee-future-prediction | Binary | 0.544 | 0.650 | -0.106 | -0.047 |
| adityakadiwal/water-potability | Binary | 0.472 | 0.610 | -0.139 | -0.034 |
| buntyshah/auto-insurance-claims-data | Binary | 0.655 | 0.753 | -0.098 | -0.030 |
| vstacknocopyright/blood-transfusion-service-center-data | Binary | 0.553 | 0.762 | -0.209 | -0.029 |
| ahsan81/food-ordering-and-delivery-app-dataset | Categorical | 0.246 | 0.400 | -0.154 | -0.021 |
| arnavsmayan/netflix-userbase-dataset | Categorical | 0.333 | 0.422 | -0.089 | -0.019 |
| receplyasolu/6k-weather-labeled-spotify-songs | Categorical | 0.166 | 0.280 | -0.114 | -0.018 |
| arnavsmayan/vehicle-manufacturing-dataset | Categorical | 0.054 | 0.111 | -0.058 | -0.009 |
| kumargh/pimaindiansdiabetescsv | Categorical | 0.111 | 0.176 | -0.065 | -0.005 |
| hansrobertson/american-companies-profits-and-benefits-from-ai | Quartile | 0.246 | 0.253 | -0.007 | -0.005 |
| jillanisofttech/brain-stroke-dataset | Binary | 0.958 | 0.959 | -0.001 | -0.002 |

*Table 7.* Performance by task type, measured as lift over majority-class baseline (mean and median) and and percentage of datasets performing at or below the baseline ($\leq$ Baseline)

| TASK TYPE | N | MEAN LIFT | MEDIAN LIFT | $\leq$ MAJ. BASELINE (%) |
|---|---|---|---|---|
| Quartile | 46 | +35.6 | +32.9 | 10.9 |
| Categorical | 55 | +10.9 | −0.3 | 54.5 |
| Binary | 64 | +2.7 | +2.5 | 46.9 |

## D. Quartile Classification May Be Artificially Easy

Quartile classification tasks discretize continuous targets into four bins based on quartile boundaries. Examining this task format reveals several properties that may inflate performance independent of genuine tabular reasoning:

*Table 8.* Pairwise $t$-tests comparing lift over majority-class baseline across task types. All comparisons are statistically significant at 0.05 significance threshold.

| COMPARISON | $t$-STATISTIC | $p$-VALUE |
|---|---|---|
| Quartile vs. Binary | 8.73 | $3.49 \times 10^{-14}$ |
| Quartile vs. Categorical | 4.62 | $1.52 \times 10^{-5}$ |
| Categorical vs. Binary | 2.25 | $2.65 \times 10^{-2}$ |

1. **Numeric leakage.** quartile classification tasks in this benchmark predict a continuous target (e.g., stock closing price) discretized into quartile bins. However, the input features often include highly correlated values. For example, predicting "Close" price when "Open," "High," "Low," and "Adj Close" are provided as features creates trivial numeric shortcuts—if Open = 603, the model can infer Close > 509 without learning any generalizable pattern.
2. **Balanced by construction.** Quartile bins guarantee 25% of samples per class *by definition*. This is a consequence of how quartiles are computed, not a property of the underlying data (which can be quite skewed or concentrated). In contrast, binary and categorical tasks exhibit natural class imbalance, making majority-class prediction a stronger baseline.
3. **Ordinal structure.** The four quartile bins (e.g., "less than X," "between X and Y," "greater than Z") have inherent ordinal relationships that language models may exploit through numeric reasoning capabilities acquired during pretraining (Shah et al., 2023), rather than learning tabular-specific representations.

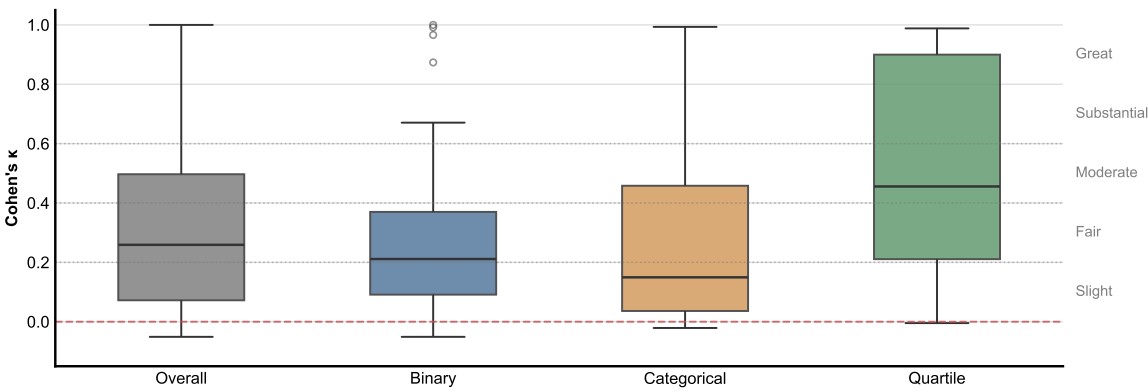

*Figure 5.* Cohen's $\kappa$ distribution by task type. Horizontal lines indicate standard interpretation thresholds.

**Complete Overlap: bitcoin-price-2014-2023**

**(a) Test Example 1: 2019-05-27**

```
Test Input
Date = 2019-05-27
Open = 8674.07
High = 8907.17
Low = 8668.71
Adj Close = 8805.78
Close = ?
```

```
T4 (3 chunks)
Date = 2019-05-27
Open = 8674.07
High = 8907.17
Low = 8668.71
Adj Close = 8805.78
Close = 8805.78
```

Label exposed in chunks 0049, 0092, 0097.

**(b) Test Example 2: 2021-04-20**

```
Test Input
Date = 2021-04-20
Open = 55681.79
High = 57062.15
Low = 53448.05
Adj Close = 56473.03
Close = ?
```

```
T4 (4 chunks)
Date = 2021-04-20
Open = 55681.79
High = 57062.15
Low = 53448.05
Adj Close = 56473.03
Close = 56473.03
```

Label exposed in chunks 0044, 0057, 0081, 0097.

*Figure 6.* Contamination in Bitcoin price prediction. Test examples appear verbatim in multiple T4 chunks with target labels exposed. Additionally, `Adj Close` $\approx$ `Close` provides a numeric shortcut.

# E. Extended Contamination Examples

**US Women's Labor Force Participation: Extended Evidence.** As described in Section 4.2, all 753 test rows from this dataset appear in T4 with matching labels. Table 9 provides the first 20 matched rows. Columns show the T4 field names (`k5`, `k618`, `inc`); corresponding test fields use longer names (`kids5`, `kids618`, `nwifeinc`). Test labels are encoded as 1/0; T4 labels as yes/no. All 753 rows exhibit exact value and label matches.

| Row | age | k5 | k618 | inc | Test Label | T4 Label | Match |
|-----|-----|----|------|------|-----------|----------|-------|
| 1 | 32 | 1 | 0 | 10.9 | 1 | yes | ✓ |
| 2 | 30 | 0 | 2 | 19.5 | 1 | yes | ✓ |
| 3 | 35 | 1 | 3 | 12.0 | 1 | yes | ✓ |
| 4 | 34 | 0 | 3 | 6.8 | 1 | yes | ✓ |
| 5 | 31 | 1 | 2 | 20.1 | 1 | yes | ✓ |
| 6 | 54 | 0 | 0 | 9.9 | 1 | yes | ✓ |
| 7 | 37 | 0 | 2 | 9.2 | 1 | yes | ✓ |
| 8 | 54 | 0 | 0 | 10.9 | 1 | yes | ✓ |
| 9 | 48 | 0 | 2 | 17.3 | 1 | yes | ✓ |
| 10 | 39 | 0 | 2 | 12.9 | 1 | yes | ✓ |
| 11 | 33 | 0 | 1 | 24.3 | 1 | yes | ✓ |
| 12 | 42 | 0 | 1 | 19.7 | 1 | yes | ✓ |
| 13 | 30 | 1 | 2 | 15.0 | 1 | yes | ✓ |
| 14 | 43 | 0 | 2 | 14.6 | 1 | yes | ✓ |
| 15 | 43 | 0 | 1 | 24.6 | 1 | yes | ✓ |
| 16 | 35 | 0 | 3 | 17.5 | 1 | yes | ✓ |
| 17 | 43 | 0 | 2 | 14.1 | 1 | yes | ✓ |
| 18 | 39 | 0 | 5 | 15.8 | 1 | yes | ✓ |
| 19 | 45 | 0 | 0 | 14.1 | 1 | yes | ✓ |
| 20 | 35 | 0 | 4 | 10.3 | 1 | yes | ✓ |

*... 733 additional matching rows omitted ...*

*Table 9.* First 20 rows of contamination evidence for `us-womens-labor-force-participation`. All 753 test rows appear in T4 (chunk-0098) with matching labels. Column names differ slightly (e.g., `kids618` $\rightarrow$ `k618`, `nwifeinc` $\rightarrow$ `inc`).

**Bitcoin Price: Extended Match Details.** Figure 6 provides detailed evidence of contamination in the Bitcoin price prediction task. Two representative test examples (2019-05-27 and 2021-04-20) each appear in multiple T4 chunks with the target `Close` value fully exposed. The first example appears in three chunks; the second in four. This demonstrates that individual Bitcoin price records are replicated extensively throughout T4, with the model encountering the same labeled examples multiple times during training.

**Other Financial Time Series.** The contamination pattern extends beyond Bitcoin to other top-performing financial datasets. Figure 7 shows representative examples from Netflix, Bitcoin-USD, Yahoo, and Tesla stock prediction tasks. Each exhibits the same pattern of test records appearing in T4 with target labels fully exposed. These five financial time series collectively account for half of the top 10 performers in Table 2, suggesting that the strong benchmark results on financial forecasting tasks are driven by data memorization rather than genuine predictive capability.

**Peloton: Extended Match Details.** Table 10 shows the distribution of the date-day association (2021-11-30 $\rightarrow$ Tuesday) across T4. All 844 records contain both the date and its corresponding day label, spanning power grids (chunk-0086), transit systems (chunks-0015, 0115), telecom networks (chunk-0144), and 21 other unrelated domains. The Peloton workout

**Financial Time Series Contamination**

**(a) Netflix Stock: 2020-03-19**

```
Test Input
Date = 2020-03-19
Open = 324.33
High = 348.51
Low = 316.82
Adj Close = 332.03
Close = ?
```

```
T4: chunk-0045
Open = 324.33
High = 348.51
Low = 316.82
Adj Close = 332.03
Close = 332.03
```

**(b) Bitcoin-USD: 2019-04-30**

```
Test Input
Date = 2019-04-30
Open = 5247.73
High = 5363.26
Low = 5224.19
Adj Close = 5350.73
Close = ?
```

```
T4: chunk-0097
Open = 5247.73
High = 5363.26
Low = 5224.19
Adj Close = 5350.73
Close = 5350.73
```

**(c) Yahoo Stock: 2018-09-11**

```
Test Input
Date = 2018-09-11
Open = 2871.57
High = 2892.52
Low = 2866.78
Adj Close = 2887.89
Close = ?
```

```
T4: chunk-0054
Open = 2871.57
High = 2892.52
Low = 2866.78
Adj Close = 2887.89
Close = 2887.89
```

**(d) Tesla Stock: 2017-06-26**

```
Test Input
Date = 2017-06-26
Open = 386.69
High = 386.95
Low = 373.10
Adj Close = 377.49
Close = ?
```

```
T4: chunk-0017
Open = 386.69
High = 386.95
Low = 373.10
Adj Close = 377.49
Close = 377.49
```

*Figure 7.* Contamination in financial time series datasets. Test examples appear in T4 with target `Close` values exposed. These datasets rank among the top 10 performers in the Tabula-8B evaluation.

*Table 10.* Task leakage evidence: Distribution of date-day associations in T4 training corpus. The mapping 2021-11-30 → Tuesday appears 844 times across 24 different T4 chunks spanning diverse domains.

| Dataset Domain | Matches | Representative Fields/Context |
|---|---|---|
| Power grid monitoring | 637 | Block, Time, Avg. Frequency, SG(MW), AG(MW), Unit_Running |
| Bus transit schedules | 91 | Transit Date, Block #, Badge #, Bus Stop ID, Latitude, Longitude |
| Transit delays | 38 | Date, Line (500–505), Location, Incident type, Min Delay, Vehicle |
| 4G network metrics | 24 | Period, Data Accessibility SR, Voice DCR, ERAB Attempts, Hour |
| Prayer times | 1 | Date, Fajr, Sunrise, Zuhr, Asr, Maghrib, Isha times |
| Meal delivery platform | 1 | TotalSupply, MealsSaved, PayingUsers, SupplyingStores |
| Other timestamped data | 52 | Various operational/monitoring systems across 18 chunks |
| **Total across 24 chunks** | **844** | |

dataset itself does not appear in T4, yet the task of predicting day-of-week from date is trivially solvable via memorization of this widely-attested association across hundreds of timestamped operational datasets.

## F. Alpaca Extended Results

*Table 11.* Accuracy (%) across for Tabula-8B, Base Llama, and Alpaca, for `bundesliga-seasons` and `udemy-courses`.

| DATASET | TASK | BASE | ALPACA | TABULA |
|---------|------|------|--------|--------|
| BUNDESLIGA | BINARY | 94.5 | 97.2 | 100.0 |
| UDEMY | CATEG. | 70.6 | 94.2 | 99.5 |

*Table 12.* Classification performance before and after excluding datasets with documented contamination: `pokemon` (×2), `peloton-data`, `us-womens-labor`, and `udemy-courses`[8]. Lift over majority-class baseline shown as mean and median across datasets. Gap indicates difference between Alpaca and Tabula-8B mean lift.

| DATASET SET | TABULA-8B | | ALPACA | | BASE LLAMA | | GAP |
|-------------|-----------|--------|--------|--------|------------|--------|-----|
| | MEAN | MEDIAN | MEAN | MEDIAN | MEAN | MEDIAN | (MEAN) |
| ALL CLASSIFICATION (N=119) | +6.5 | +0.0 | +1.6 | -1.5 | -9.4 | -5.4 | +4.9 |
| EXCL. CONTAMINATED (N=114) | +4.3 | -0.1 | +0.3 | -1.7 | -10.4 | -5.8 | +4.1 |
| CHANGE | -2.2 | -0.1 | -1.3 | -0.2 | -1.0 | -0.4 | -0.8 |

*Table 13.* Accuracy scores for Tabula-8B, Llama-3-8B, and Alpaca variants across 165 tabular datasets. Alpaca is Llama-3-8B fine-tuned on the Alpaca instruction dataset, while Alpaca+Q is further trained with quartile regression data. Majority baseline predicts the most frequent class. Empty cells indicate tasks used to train Alpaca+Q.

| Dataset | Type | Majority | Tabula-8B | Llama-3-8B | Alpaca | Alpaca+Q |
|---------|------|----------|-----------|------------|--------|----------|
| aakashjoshi123/exercise-and-fitness-metrics-dataset | Quartile | 0.280 | 0.577 | 0.064 | 0.299 | |
| aakashjoshi123/spotify-top-hits-data | Categorical | 0.757 | 0.721 | 0.708 | 0.720 | 0.729 |
| abcsds/pokemon | Categorical | 0.140 | 0.990 | 0.677 | 0.882 | 0.968 |
| adityakadiwal/water-potability | Binary | 0.610 | 0.472 | 0.498 | 0.519 | 0.557 |
| agirlcoding/all-space-missions-from-1957 | Categorical | 0.903 | 0.904 | 0.833 | 0.876 | 0.899 |
| ahsan81/food-ordering-and-delivery-app-dataset | Categorical | 0.400 | 0.246 | 0.248 | 0.263 | 0.275 |
| ahsan81/superstore-marketing-campaign-dataset | Binary | 0.849 | 0.730 | 0.663 | 0.778 | 0.634 |
| akshaydattatraykhare/diabetes-dataset | Binary | 0.651 | 0.707 | 0.674 | 0.694 | 0.727 |
| alexisbcook/pakistan-intellectual-capital | Categorical | 0.650 | 0.649 | 0.216 | 0.281 | 0.024 |
| alirezachahardoli/bank-personal-loan-1 | Binary | 0.899 | 0.843 | 0.879 | 0.865 | 0.847 |
| altruistdelhite04/gold-price-data | Quartile | 0.265 | 0.742 | 0.121 | 0.289 | |
| amirhosseinmirzaie/countries-life-expectancy | Quartile | 0.266 | 0.756 | 0.273 | 0.353 | |
| amirhosseinmirzaie/pistachio-types-detection | Binary | 0.575 | 0.668 | 0.417 | 0.519 | 0.590 |
| ananthr1/weather-prediction | Categorical | 0.438 | 0.774 | 0.001 | 0.604 | 0.668 |
| andrewmvd/fetal-health-classification | Categorical | 0.791 | 0.742 | 0.733 | 0.662 | 0.514 |
| andrewmvd/udemy-courses | Categorical | 0.316 | 0.995 | 0.706 | 0.942 | 0.988 |
| arashnic/time-series-forecasting-with-yahoo-stock-price | Quartile | 0.257 | 0.988 | 0.244 | 0.409 | 0.731 |
| arnabchaki/data-science-salaries-2023 | Quartile | 0.273 | 0.947 | 0.001 | 0.556 | |
| arnabchaki/indian-restaurants-2023 | Quartile | 0.365 | 0.276 | 0.128 | 0.303 | 0.300 |
| arnavsmayan/netflix-userbase-dataset | Categorical | 0.422 | 0.333 | 0.344 | 0.378 | 0.363 |
| arnavsmayan/vehicle-manufacturing-dataset | Categorical | 0.111 | 0.054 | 0.066 | 0.083 | 0.077 |
| arslanr369/bitcoin-price-2014-2023 | Quartile | 0.263 | 0.991 | 0.175 | 0.326 | 0.829 |
| arslanr369/roblox-stock-pricing-2021-2023 | Quartile | 0.250 | 0.969 | 0.012 | 0.355 | |
| ashishkumarjayswal/diabetes-dataset | Binary | 0.651 | 0.677 | 0.648 | 0.669 | 0.712 |
| ashishkumarjayswal/loanamount-approval | Binary | 0.687 | 0.780 | 0.169 | 0.596 | 0.630 |
| ashishkumarjayswal/movies-updated-data | Categorical | 0.342 | 0.637 | 0.479 | 0.580 | 0.554 |
| atharvaingle/crop-recommendation-dataset | Categorical | 0.052 | 0.223 | 0.000 | 0.057 | 0.101 |
| awaiskaggler/insurance-csv | Quartile | 0.265 | 0.624 | 0.068 | 0.293 | |
| azminetoushikwasi/-lionel-messi-all-club-goals | Categorical | 0.619 | 0.445 | 0.214 | 0.385 | 0.284 |
| barun2104/telecom-churn | Binary | 0.863 | 0.851 | 0.748 | 0.798 | 0.810 |
| bhanupratapbiswas/bollywood-actress-name-and-movie-list | Categorical | 0.571 | 0.503 | 0.246 | 0.449 | 0.404 |
| bhanupratapbiswas/fashion-products | Categorical | 0.351 | 0.345 | 0.311 | 0.336 | 0.338 |
| bhanupratapbiswas/ipl-dataset-2008-2016 | Categorical | 0.139 | 0.939 | 0.624 | 0.593 | 0.671 |

---

[8]The `udemy-courses` task classifies courses into four subjects based on `course_title`; likely a text classification task solvable through pretrained LLMs.

| Dataset | Type | Majority | Tabula-8B | Llama-3-8B | Alpaca | Alpaca+Q |
|---|---|---|---|---|---|---|
| bhanupratapbiswas/uber-data-analysis | Categorical | 0.932 | 0.845 | 0.922 | 0.922 | 0.858 |
| bhanupratapbiswas/world-top-billionaires | Quartile | 0.267 | 0.557 | 0.245 | 0.322 | 0.456 |
| bharath011/heart-disease-classification-dataset | Binary | 0.623 | 0.612 | 0.435 | 0.668 | 0.666 |
| bhavkaur/hotel-guests-dataset | Categorical | 0.867 | 0.606 | 0.096 | 0.508 | 0.697 |
| bhavkaur/simplified-titanic-dataset | Binary | 0.749 | 0.600 | 0.523 | 0.661 | 0.555 |
| blastchar/telco-customer-churn | Binary | 0.731 | 0.691 | 0.278 | 0.722 | 0.718 |
| bretmathyer/telemedicine-used | Binary | 0.512 | 0.562 | 0.488 | 0.482 | 0.501 |
| buntyshah/auto-insurance-claims-data | Binary | 0.753 | 0.655 | 0.240 | 0.736 | 0.751 |
| burak3ergun/loan-data-set | Binary | 0.687 | 0.792 | 0.436 | 0.567 | 0.648 |
| carolzhangdc/imdb-5000-movie-dataset | Quartile | 0.288 | 0.577 | 0.233 | 0.259 | 0.518 |
| chirin/africa-economic-banking-and-systemic-crisis-data | Binary | 0.910 | 0.934 | 0.197 | 0.836 | 0.846 |
| christinestevens/cstevens-peloton-data | Categorical | 0.212 | 0.994 | 0.006 | 0.212 | 0.942 |
| cpluzshrijayan/milkquality | Categorical | 0.404 | 0.392 | 0.394 | 0.482 | 0.481 |
| crxxom/manhwa-dataset | Categorical | 0.636 | 0.886 | 0.596 | 0.618 | 0.838 |
| dansbecker/aer-credit-card-data | Binary | 0.783 | 0.725 | 0.648 | 0.658 | 0.624 |
| deependraverma13/diabetes-healthcare-comprehensive-dataset | Binary | 0.651 | 0.688 | 0.678 | 0.695 | 0.719 |
| desalegngeb/german-fintech-companies | Categorical | 0.362 | 0.832 | 0.046 | 0.527 | 0.687 |
| dileep070/heart-disease-prediction-using-logistic-regression | Binary | 0.855 | 0.782 | 0.670 | 0.726 | 0.642 |
| dsfelix/us-stores-sales | Quartile | 0.271 | 0.674 | 0.123 | 0.457 | |
| elakiricoder/gender-classification-dataset | Binary | 0.521 | 0.732 | 0.518 | 0.844 | 0.765 |
| fedesoriano/hepatitis-c-dataset | Categorical | 0.867 | 0.855 | 0.867 | 0.868 | 0.857 |
| fedesoriano/stroke-prediction-dataset | Binary | 0.949 | 0.949 | 0.805 | 0.864 | 0.849 |
| gabrielsantello/cars-purchase-decision-dataset | Binary | 0.598 | 0.602 | 0.650 | 0.612 | 0.605 |
| geomack/spotifyclassification | Binary | 0.510 | 0.576 | 0.174 | 0.518 | 0.555 |
| gyanprakashkushwaha/laptop-price-prediction-cleaned-dataset | Quartile | 0.260 | 0.506 | 0.175 | 0.315 | |
| hansrobertson/american-companies-profits-and-benefits-from-ai | Quartile | 0.253 | 0.246 | 0.048 | 0.280 | 0.256 |
| harishkumardatalab/medical-insurance-price-prediction | Quartile | 0.254 | 0.659 | 0.036 | 0.324 | 0.834 |
| harshitshankhdhar/imdb-dataset-of-top-1000-movies-and-tv-shows | Quartile | 0.322 | 0.406 | 0.291 | 0.338 | |
| hashemi221022/bank-loans | Binary | 0.919 | 0.877 | 0.914 | 0.901 | 0.868 |
| hashemi221022/diabetes | Binary | 0.651 | 0.691 | 0.682 | 0.664 | 0.711 |
| hawkingcr/airbnb-for-boston-with-fraud-detection | Binary | 0.792 | 0.749 | 0.725 | 0.682 | 0.618 |
| hemanthhari/psycological-effects-of-covid | Quartile | 0.293 | 0.421 | 0.318 | 0.302 | |
| hesh97/titanicdataset-traincsv | Binary | 0.616 | 0.763 | 0.633 | 0.676 | 0.710 |
| iamsumat/spotify-top-2000s-mega-dataset | Quartile | 0.256 | 0.449 | 0.085 | 0.288 | |
| iqmansingh/company-employee-dataset | Categorical | 0.150 | 0.537 | 0.085 | 0.101 | 0.508 |
| ishadss/productivity-prediction-of-garment-employees | Quartile | 0.254 | 0.395 | 0.079 | 0.292 | 0.347 |
| jainilcoder/netflix-stock-price-prediction | Quartile | 0.251 | 0.982 | 0.000 | 0.388 | 0.745 |
| jillanisofttech/brain-stroke-dataset | Binary | 0.959 | 0.958 | 0.834 | 0.879 | 0.858 |
| kabure/german-credit-data-with-risk | Binary | 0.700 | 0.573 | 0.477 | 0.646 | 0.675 |
| kandij/diabetes-dataset | Binary | 0.651 | 0.698 | 0.677 | 0.680 | 0.721 |
| kanths028/usa-housing | Quartile | 0.262 | 0.276 | 0.031 | 0.258 | 0.370 |
| kingabzpro/cosmetics-datasets | Categorical | 0.217 | 0.810 | 0.213 | 0.710 | 0.768 |
| kreeshrajani/human-stress-prediction | Binary | 0.538 | 0.566 | 0.555 | 0.613 | 0.613 |
| kumargh/pimaindiansdiabetescsv | Categorical | 0.176 | 0.111 | 0.100 | 0.097 | 0.108 |
| larsen0966/student-performance-data-set | Categorical | 0.160 | 0.291 | 0.305 | 0.336 | 0.390 |
| lightonkalumba/us-womens-labor-force-participation | Binary | 0.568 | 0.996 | 0.534 | 0.703 | 0.892 |
| mahnazarjmand/bank-personal-loan | Binary | 0.916 | 0.884 | 0.904 | 0.887 | 0.872 |
| maryalebron/life-expectancy-data | Categorical | 0.038 | 0.028 | 0.033 | 0.021 | 0.023 |
| maryammanoochehry/bank-personal-loan | Binary | 0.897 | 0.870 | 0.870 | 0.864 | 0.875 |
| mathchi/diabetes-data-set | Binary | 0.651 | 0.691 | 0.697 | 0.703 | 0.720 |
| mayankpatel14/second-hand-used-cars-data-set-linear-regression | Quartile | 0.250 | 0.262 | 0.008 | 0.264 | 0.282 |
| mayurdalvi/simple-linear-regression-placement-data | Binary | 0.511 | 0.507 | 0.480 | 0.486 | 0.501 |
| mayuriawati/bangalore-chain-restaurants-ratings-and-reviews | Categorical | 0.157 | 0.885 | 0.618 | 0.818 | 0.883 |
| mazlumi/ielts-writing-scored-essays-dataset | Categorical | 0.178 | 0.218 | 0.227 | 0.199 | 0.179 |
| mfaisalqureshi/spam-email | Binary | 0.871 | 0.905 | 0.796 | 0.846 | 0.749 |
| mirichoi0218/insurance | Quartile | 0.255 | 0.602 | 0.097 | 0.339 | |

| Dataset | Type | Majority | Tabula-8B | Llama-3-8B | Alpaca | Alpaca+Q |
|---|---|---|---|---|---|---|
| muhammadtsabitulazmi/liga-1-indonesia-player-dataset | Categorical | 0.160 | 0.206 | 0.002 | 0.130 | 0.136 |
| nancyalaswad90/review | Binary | 0.651 | 0.680 | 0.671 | 0.685 | 0.723 |
| naveenkumar20bps1137/predict-students-dropout-and-academic-success | Categorical | 0.167 | 0.082 | 0.109 | 0.088 | 0.082 |
| nikhil1e9/netflix-stock-price | Quartile | 0.255 | 0.990 | 0.188 | 0.360 | 0.791 |
| ninzaami/loan-predication | Binary | 0.687 | 0.785 | 0.241 | 0.622 | 0.677 |
| noordeen/insurance-premium-prediction | Quartile | 0.255 | 0.565 | 0.059 | 0.315 | |
| oles04/bundesliga-seasons | Binary | 0.588 | 1.000 | 0.945 | 0.972 | 0.978 |
| oles04/top-leagues-player | Quartile | 0.273 | 0.678 | 0.045 | 0.290 | |
| patelprashant/employee-attrition | Binary | 0.837 | 0.829 | 0.803 | 0.836 | 0.822 |
| pavansubhasht/ibm-hr-analytics-attrition-dataset | Binary | 0.836 | 0.832 | 0.689 | 0.833 | 0.817 |
| phangud/spamcsv | Binary | 0.864 | 0.896 | 0.726 | 0.912 | 0.761 |
| prevek18/ames-housing-dataset | Quartile | 0.271 | 0.582 | 0.304 | 0.291 | |
| primaryobjects/voicegender | Binary | 0.501 | 0.630 | 0.269 | 0.551 | 0.598 |
| prkhrawsthi/bitcoin-usd-daily-price-with-volume-2015-2023 | Quartile | 0.283 | 0.990 | 0.098 | 0.323 | 0.665 |
| raddar/icr-integer-data | Binary | 0.825 | 0.731 | 0.762 | 0.778 | 0.363 |
| rajyellow46/wine-quality | Categorical | 0.437 | 0.389 | 0.362 | 0.368 | 0.333 |
| ravibarnawal/mutual-funds-india-detailed | Categorical | 0.299 | 0.236 | 0.240 | 0.259 | 0.291 |
| receplyasolu/6k-weather-labeled-spotify-songs | Categorical | 0.280 | 0.166 | 0.000 | 0.145 | 0.127 |
| redwankarimsony/heart-disease-data | Categorical | 0.447 | 0.359 | 0.228 | 0.321 | 0.310 |
| reihanenamdari/breast-cancer | Quartile | 0.273 | 0.296 | 0.260 | 0.271 | 0.298 |
| rishikeshkonapure/hr-analytics-prediction | Binary | 0.839 | 0.826 | 0.800 | 0.840 | 0.814 |
| rkiattisak/student-performance-in-mathematics | Quartile | 0.266 | 0.543 | 0.200 | 0.496 | 0.569 |
| rounakbanik/pokemon | Binary | 0.913 | 0.981 | 0.964 | 0.966 | 0.975 |
| rpaguirre/tesla-stock-price | Quartile | 0.261 | 0.985 | 0.076 | 0.330 | 0.816 |
| rtatman/chocolate-bar-ratings | Categorical | 0.221 | 0.163 | 0.220 | 0.200 | 0.171 |
| ruchi798/student-feedback-survey-responses | Categorical | 0.111 | 0.104 | 0.098 | 0.104 | 0.099 |
| ruchi798/tv-shows-on-netflix-prime-video-hulu-and-disney | Categorical | 0.395 | 0.354 | 0.202 | 0.223 | 0.238 |
| sabasaeed1953/stock-prices-of-2023 | Quartile | 0.250 | 0.980 | 0.070 | 0.366 | 0.623 |
| saloni1712/chatgpt-app-reviews | Categorical | 0.520 | 0.681 | 0.660 | 0.642 | 0.690 |
| sanjanchaudhari/bankloan | Binary | 0.629 | 0.610 | 0.330 | 0.612 | 0.582 |
| sanjanchaudhari/netflix-dataset | Categorical | 0.325 | 0.534 | 0.428 | 0.452 | 0.565 |
| sanjanchaudhari/user-behavior-on-instagram | Binary | 0.507 | 0.583 | 0.492 | 0.487 | 0.488 |
| saunakghosh/nba-players-dataset | Categorical | 0.750 | 0.772 | 0.556 | 0.651 | 0.601 |
| saurabh00007/diabetescsv | Binary | 0.651 | 0.714 | 0.664 | 0.698 | 0.737 |
| sbhatti/financial-sentiment-analysis | Categorical | 0.542 | 0.641 | 0.099 | 0.694 | 0.756 |
| shashankshukla123123/marketing-campaign | Binary | 0.844 | 0.771 | 0.565 | 0.819 | 0.690 |
| shivamb/disney-movies-and-tv-shows | Binary | 0.727 | 0.986 | 0.639 | 0.895 | 0.938 |
| shivamb/hm-stores-dataset | Categorical | 0.557 | 0.483 | 0.165 | 0.548 | 0.472 |
| shreyanshverma27/imdb-horror-chilling-movie-dataset | Quartile | 0.293 | 0.537 | 0.145 | 0.299 | 0.402 |
| shreyapurohit/anime-data | Quartile | 0.269 | 0.777 | 0.009 | 0.374 | |
| shroukgomaa/babies-food-ingredients | Categorical | 0.339 | 0.293 | 0.300 | 0.286 | 0.343 |
| shubhamgupta012/titanic-dataset | Binary | 0.618 | 0.754 | 0.668 | 0.642 | 0.690 |
| siddharthss/crop-recommendation-dataset | Categorical | 0.057 | 0.212 | 0.003 | 0.057 | 0.080 |
| sidhus/crab-age-prediction | Categorical | 0.153 | 0.109 | 0.102 | 0.117 | 0.097 |
| sudarshan6561/ipl-2023 | Categorical | 0.357 | 0.637 | 0.060 | 0.521 | 0.669 |
| suraj520/dairy-goods-sales-dataset | Quartile | 0.268 | 0.631 | 0.244 | 0.286 | |
| surajjha101/stores-area-and-sales-data | Quartile | 0.250 | 0.250 | 0.007 | 0.305 | 0.276 |
| surajjha101/top-youtube-channels-data | Categorical | 0.241 | 0.504 | 0.305 | 0.445 | 0.504 |
| swathiunnikrishnan/amazon-consumer-behaviour-dataset | Quartile | 0.331 | 0.286 | 0.221 | 0.297 | 0.324 |
| tarkkaanko/amazon | Categorical | 0.775 | 0.823 | 0.798 | 0.762 | 0.786 |
| team-ai/spam-text-message-classification | Binary | 0.863 | 0.897 | 0.739 | 0.903 | 0.761 |
| teertha/ushealthinsurancedataset | Quartile | 0.256 | 0.617 | 0.046 | 0.331 | |
| tejashvi14/employee-future-prediction | Binary | 0.650 | 0.544 | 0.540 | 0.545 | 0.566 |
| tejashvi14/engineering-placements-prediction | Binary | 0.556 | 0.631 | 0.666 | 0.661 | 0.694 |
| thedevastator/cancer-patients-and-air-pollution-a-new-link | Categorical | 0.077 | 0.057 | 0.053 | 0.069 | 0.055 |
| thedevastator/employee-attrition-and-factors | Binary | 0.833 | 0.830 | 0.724 | 0.834 | 0.818 |
| thedevastator/higher-education-predictors-of-student-retention | Binary | 0.867 | 0.861 | 0.835 | 0.793 | 0.779 |
| therealsampat/predict-movie-success-rate | Categorical | 0.821 | 0.806 | 0.772 | 0.797 | 0.781 |
| timoboz/tesla-stock-data-from-2010-to-2020 | Quartile | 0.268 | 0.988 | 0.000 | 0.399 | 0.765 |
| uciml/indian-liver-patient-records | Binary | 0.714 | 0.616 | 0.511 | 0.583 | 0.554 |

| Dataset | Type | Majority | Tabula-8B | Llama-3-8B | Alpaca | Alpaca+Q |
|---|---|---|---|---|---|---|
| uciml/mushroom-classification | Binary | 0.511 | 0.767 | 0.097 | 0.557 | 0.660 |
| uciml/pima-indians-diabetes-database | Binary | 0.651 | 0.684 | 0.673 | 0.681 | 0.728 |
| uciml/red-wine-quality-cortez-et-al-2009 | Categorical | 0.420 | 0.421 | 0.291 | 0.393 | 0.405 |
| varpit94/tesla-stock-data-updated-till-28jun2021 | Quartile | 0.271 | 0.980 | 0.005 | 0.438 | 0.879 |
| vedavyasv/usa-housing | Quartile | 0.257 | 0.301 | 0.088 | 0.246 | 0.326 |
| vijayvvenkitesh/microsoft-stock-time-series-analysis | Quartile | 0.267 | 0.985 | 0.004 | 0.356 | 0.756 |
| vikramamin/customer-churn-decision-tree-and-random-forest | Binary | 0.725 | 0.692 | 0.289 | 0.720 | 0.716 |
| vikramamin/time-series-forecasting-using-prophet-in-r | Quartile | 0.253 | 0.365 | 0.000 | 0.337 | 0.445 |
| vstacknocopyright/blood-transfusion-service-center-data | Binary | 0.762 | 0.553 | 0.683 | 0.662 | 0.627 |
| warcoder/earthquake-dataset | Quartile | 0.303 | 0.861 | 0.144 | 0.456 | |
| wearefuture01/hepatitis-c-prediction | Categorical | 0.867 | 0.863 | 0.857 | 0.865 | 0.841 |
| whenamancodes/predict-diabities | Binary | 0.651 | 0.681 | 0.673 | 0.699 | 0.719 |
| whenamancodes/students-performance-in-exams | Quartile | 0.275 | 0.573 | 0.099 | 0.501 | |
| yasserh/titanic-dataset | Binary | 0.616 | 0.786 | 0.632 | 0.670 | 0.712 |
| yasserh/wine-quality-dataset | Categorical | 0.427 | 0.427 | 0.375 | 0.389 | 0.333 |
| ybifoundation/food-app-business | Quartile | 0.380 | 0.276 | 0.186 | 0.315 | 0.236 |

# G. Evaluation Gaps Are Endemic Across TLM Research

A natural concern with our analysis is that it focuses on a single case study. To address this, we systematically examined the evaluation methodology of five other influential TLM papers. Our analysis provides further evidence that the gaps we identify in Tabula-8B are endemic patterns across the TLM literature.

We selected five papers representing the breadth of recent TLM research:
1. **Fang et al. (2024) Survey**: A comprehensive survey that consolidates evaluation practices and recommends benchmark datasets.
2. **TabLLM** (Hegselmann et al., 2023): This work introduced few-shot classification via serialization and prompting, evaluating on 12 datasets.
3. **UniPredict** (Wang et al., 2023): A paper training GPT-2 on 169 Kaggle datasets, claiming universal classification capabilities.
4. **GTL** (Sun et al., 2024): This work scales generative tabular learning to 972 datasets.
5. **CARTE** (Kim et al., 2024): While not strictly being a general TLM, this work proposes context-aware representations using graph-attentional for tabular transfer learning, evaluating on 51 datasets.

We evaluated each paper against the same criteria applied to our Tabula-8B analysis.

## G.1. Findings

### G.1.1. UNIVERSAL ABSENCE OF TRIVIAL BASELINES

The most striking finding is the complete absence of majority-class baseline reporting across all five papers. None report the accuracy achievable by simply predicting the most frequent class, and consequently none report lift over this baseline. Additionally, none report chance-corrected metrics such as Cohen's $\kappa$. This omission is consequential. UniPredict, for instance, reports only raw accuracy across 169 datasets with no chance correction. The paper claims "a notable increase in absolute accuracy of 2.2% when compared to XGBoost," but without majority-baseline context, it is impossible to determine whether this represents meaningful lift or noise on potentially imbalanced datasets. TabLLM uses AUC as its primary metric, which is more robust to class imbalance than accuracy, but never discusses class distributions in its evaluation datasets.

### G.1.2. INCOMPLETE TASK-TYPE STRATIFICATION

While some papers separate classification from regression, none separate binary from multiclass classification, and none conduct statistical tests for heterogeneity across task types. UniPredict is the most problematic case, reporting only aggregate accuracy across all 169 datasets. The paper notes that "for continuous numerical targets (e.g., regression), the categories are defined by their quantiles," converting regression to classification, but provides no separate analysis of these converted tasks versus natural classification problems. In contrast, CARTE provides the best stratification among the papers we examined, with separate learning curves for 40 regression datasets and 11 classification datasets. However, even CARTE does not separate binary from multiclass classification or test whether the method's advantage varies systematically by task type.

### G.1.3. CONTAMINATION ANALYSIS IS UNIVERSALLY ABSENT

All five papers rely on publicly available datasets from UCI, Kaggle, and OpenML that are almost certainly present in LLM training corpora, yet none conduct contamination analysis. This is particularly concerning given recent evidence that LLMs have memorized canonical tabular benchmarks (Bordt et al., 2024; Carlini et al., 2023).

The Fang et al. (2024) survey implicitly acknowledges the problem, noting that "GPT models were trained on a significant amount of web data and thus, probably exposed to more HTML and XML formats when interpreting tables," but never connects this observation to evaluation validity. The survey recommends a "combo of 9 datasets for benchmark" that includes UCI classics such as Adult (Income), Diabetes (Pima Indians), and Heart Disease. Bordt et al. (2024) demonstrated that GPT-3.5 and GPT-4 have memorized many popular tabular datasets verbatim. For example, GPT-4 can consistently generate the entire Iris and Wine datasets from the UCI machine learning repository. The authors found that LLMs perform better on datasets seen during training, indicating that memorization leads to overfitting.

TabLLM's evaluation suite also consists almost entirely of high-contamination-risk datasets, including Adult Income, German Credit, California Housing, and several other UCI classics. UniPredict evaluates on 169 Kaggle datasets including multiple versions of Pima Indians Diabetes, Titanic, and Wine Quality, with no contamination discussion. CARTE uses YAGO3 (built from Wikidata and Wikipedia) for pretraining, then evaluates on datasets containing entities that likely

appear in YAGO3, such as company employees and movies, without overlap analysis. This concern is supported by Silvestri et al. (2025). They found that contamination effects emerge for datasets containing strong semantic cues, such as meaningful column names or interpretable value categories. When such cues are removed or randomized, performance drops to near-random levels. This finding suggests that LLMs' apparent competence on tabular reasoning tasks may reflect memorization of publicly available datasets rather than genuine generalization.

The only TLM paper we identified that explicitly addresses contamination is Tabula-8B (Gardner et al., 2024), which states "Given that T4 consists of 4M tables sourced from public data sources (Common Crawl, Github) and that our evaluations are also comprised of public benchmarks, we investigate the extent and possible impact of data contamination." TabPFN (Hollmann et al., 2023) avoids the problem entirely through synthetic pretraining on procedurally-generated datasets.

### G.1.4. INSTRUCTION-TUNING CONFOUNDED WITH TABULAR LEARNING

Next, existing works inadequately control for instruction-following capability in TLM evaluation, making it unclear whether performance gains stem from genuine tabular reasoning or general language model capabilities. TabLLM uses T0 and GPT-3 (`text-davinci-002`), both of which are instruction-tuned. However, they do not include comparisons with base models (e.g., T5) that lack instruction-tuning or additional fine-tuning. Thus one is no able to distinguish whether performance comes from prior knowledge of specific datasets, general instruction-following capability, semantic understanding of feature names, or actual tabular reasoning ability.

UniPredict fine-tunes GPT-2 and uses GPT-3.5 for metadata reformatting but provides no ablation separating these capabilities. No comparison with raw GPT-2 or other instruction-tuned models without tabular exposure is included.

The GTL paper provides the most informative (though still partial) control. Raw LLaMA-2 achieves approximately $\approx 0.50$ AUROC (chance level) on classification tasks, while instruction-tuned Phi-3 models achieve 0.58–0.63 AUROC without GTL training. This at least demonstrates that base models do not perform tabular classification well, but does not fully separate instruction-following from tabular pattern learning in the trained models.

### G.1.5. REPRODUCIBILITY

Table 14 summarizes reproducibility practices across the five papers. CARTE demonstrates best practices for reproducibility among the papers examined, releasing both code (BSD-3-Clause license) and datasets on HuggingFace, with results reported across 10 random train/test splits including standard deviations. TabLLM releases code but not raw predictions, and healthcare dataset code is withheld "due to privacy concerns." Alternatively, UniPredict stands out negatively with no code, no predictions, no standard deviations across runs, no confidence intervals, and no statistical significance tests. The paper evaluates on 169 datasets but provides no way to verify results or understand variance.

*Table 14.* Reproducibility practices across five major TLM papers. No paper releases raw predictions enabling independent verification.

| PAPER | CODE | PREDS. | DATA |
|---|---|---|---|
| FANG ET AL. SURVEY | ✓ | ✗ | ✓ |
| TABLLM | ✓ | ✗ | ✓ |
| UNIPREDICT | ✗ | ✗ | ✓ |
| GTL SCALING | PARTIAL | ✗ | ✓ |
| CARTE | ✓ | ✗ | ✓ |

### G.2. Summary

Table 15 consolidates our findings across all evaluation dimensions. TLM papers exhibit similar methodological gaps. This evidence strongly supports the conclusion that the issues we identify in our Tabula-8B case study are representative of endemic field-wide issues rather than isolated oversights. These findings do not diminish the contributions of individual papers, many of which have advanced the field in important ways. Rather, they underscore the need for field-wide adoption of the evaluation standards we propose in Section 5 (or at least initiate a broader discussion around what the standards should be).

*Table 15.* Evaluation methodology scorecard across four other major TLM approaches. ✓ = adequately addressed; ∼ = partially addressed; ✗ = not addressed. All four papers exhibit the same gaps identified in our Tabula-8B case study, indicating that these issues are endemic to the TLM literature rather than specific to any single work.

| DIMENSION | TABLLM | UNIPREDICT | GTL | CARTE |
|---|---|---|---|---|
| MAJORITY-CLASS BASELINE | ✗ | ✗ | ✗ | ✗ |
| CHANCE-CORRECTED METRICS | ✗ | ✗ | ✗ | ✗ |
| BINARY/QUARTILE/MULTICLASS SPLIT | ✗ | ✗ | ✗ | ∼ |
| TASK HETEROGENEITY ANALYSIS | ✗ | ✗ | ✗ | ✗ |
| CONTAMINATION ANALYSIS | ✗ | ✗ | ✗ | ✗ |
| INSTRUCTION-TUNING CONTROLS | ✗ | ✗ | ∼ | ✗ |
| GAPS IDENTIFIED | 6/6 | 6/6 | 5/6 | 5/6 |

## G.3. Extending the Framework for Closed-Source TLMs

Several recent TLMs ship without training-corpus or weight disclosure (e.g., commercial systems exposed only through an API), which limits direct application of our recommendations. We sketch how the framework degrades for this setting; workarounds below are offered as in-principle approaches rather than validated procedures.

**Fully transferable.** R1 (majority-class baselines and chance-corrected metrics), R2 (task-type stratification), R5 (task auditing for tabular-reasoning requirements), R6 (instruction-tuned-baseline parity tests), and R7 (community scrutiny) all operate on per-dataset evaluation outputs and do not require model internals. A closed-source TLM that does not substantially outperform an open instruction-tuned baseline under R6 leaves the central claim that tabular pretraining drives performance weakly supported.

**Partially transferable.** R4 (contamination beyond row-level deduplication) can no longer search the training corpus directly, but three behavioral workarounds are available in principle. First, *perturbation tests*, where memorized canonical datasets are altered to break surface cues and accuracy drops are interpreted as evidence of memorization. Second, *time-based holdouts*, where evaluation datasets are constructed after the model's reported training-data cutoff. Third, *verbatim-extraction (canary) probes* of the kind introduced by Bordt et al. (2024), who show that GPT-3.5 and GPT-4 reproduce canonical tabular datasets such as Iris and Wine verbatim; querying a closed-source endpoint for suspected near-verbatim records directly tests for the same contamination. Each probe upper-bounds rather than measures contamination and should be treated as suggestive.

**Not applicable.** R3 (release evaluation code) cannot be enforced on closed-source authors; the recommendation instead becomes a request for evaluator-side code release so that third parties can rerun the same protocol on whichever endpoints they choose.

The net effect is that closed-source TLMs can still be evaluated rigorously on five of seven recommendations, with R4 weakened but not eliminated and only R3 fully out of reach.

## G.4. Extending the Framework for Table-Synthesis Methods

A separate but adjacent line of work uses language models to *generate* synthetic tabular data rather than classify it. Representative examples include GReaT (Borisov et al., 2023), which fine-tunes GPT-style models to produce realistic tabular records; TapTap (Zhang et al., 2023), which couples generative table pre-training with downstream prediction; and Tabby (Cromp et al., 2026), which augments the Transformer architecture with column-specific Mixture-of-Experts heads for structured-data synthesis. We exclude these methods from our main scorecard for two reasons. First, our focus is the validity of *predictive* TLM evaluation, which is the regime where reported gains have driven the most adoption claims. Second, synthesis benchmarks rely on a different set of evaluation primitives, namely train-on-synthetic-test-on-real (TSTR) utility, fidelity metrics, near-duplicate detection, and privacy leakage, on which the six dimensions of Table 15 map awkwardly or not at all.

That said, two of our recommendations transfer by analogy. R1 (baselines) applies if "majority-class" is replaced with a trivial generator such as training-data resampling. R4 (contamination beyond row-level deduplication) applies more directly. Synthesis benchmarks frequently rely on canonical small datasets, and Bordt et al. (2024) show that LLMs memorize such datasets verbatim, so we would expect TSTR utility on these datasets to be inflated by leakage rather than reflect learned distributional fidelity. A full synthesis-side audit using adapted dimensions is a natural follow-up to our work but is out of scope for the present analysis.

