# OpenReview forum: "The Illusion of Generalization in Tabular Language Models"
_ICML.cc/2026/Conference — ICML 2026 regular_

### Official Review · Reviewer_3TbE · 2026-03-06

**Soundness:** 3
**Presentation:** 3
**Significance:** 3
**Originality:** 3
**Overall Recommendation:** 4
**Confidence:** 4

**Summary:**

This paper takes a systematic second look at claims that Table Language Models (TLMs) achieve “emergent generalization” on table prediction tasks. Focusing on the Tabula-8B model, the authors conducted a thorough empirical study across 165 datasets from the UniPredict benchmark. Their research uncovered three core findings: First, TLMs barely outperformed majority-class baselines on binary and categorical classification tasks, their impressive overall performance was entirely driven by quartile classification tasks. Second, the top-performing datasets suffered from widespread data contamination, including full train-test overlap and task-level data leaks that evaded standard deduplication. Finally, the authors showed that even models that never saw any table data could recover 92.2% of standard classification performance. The paper concludes convincingly that the generalization capabilities claimed by TLMs are more evaluation artifacts than genuine table reasoning skills.

**Compliance With Llm Reviewing Policy:**

Affirmed.

**Final Justification:**

Although some limitations remain, the paper's insights into systemic evaluation flaws make it a valuable contribution worthy of a Weak Accept.

**Key Questions For Authors:**

Please refer the **Weaknesses** section :-)

**Limitations:**

yes

**Strengths And Weaknesses:**

### Strengths
1. As LLMs become more widely used with table data, this paper is timely and incredibly sharp in calling out systemic flaws in current TLM evaluation systems. It plays a crucial “whistleblower” role in correcting blind optimism in the field and guiding future research.

2. The authors introduced majority-class prediction as a minimum standard baseline, effectively exposing the class imbalance illusion behind high accuracy. They also ran ablation experiments with Alpaca to peel apart the contributions of “instruction-following ability” versus “table knowledge” to model performance.

3. This paper didn’t stop at simple row-level deduplication; it dug deep to uncover three distinct forms of contamination.

4. The paper isn’t just critical, it wraps up with 7 specific, actionable evaluation recommendations covering baseline comparisons, task stratification, code open-sourcing, and contamination auditing. This adds significant academic value and constructiveness.


### Weaknesses
From a strict perspective, here are the weaknesses:
1. While Appendix G reviews evaluation methods from other papers, the core empirical analysis relies almost entirely on a single model (Tabula-8B) and a subset of UniPredict datasets. Additionally, the evaluation is limited to classification tasks, excluding other TLM capabilities like imputation, generation, or multi-table reasoning.

2. Due to computational constraints, the authors used targeted pattern searches for contamination in the T4 corpus instead of exhaustive corpus-wide matching. This lets them confirm contamination exists but prevents them from providing an accurate upper bound on the contamination rate.

---

> ### Author Rebuttal · Authors · 2026-03-30
>
> We thank the reviewer for the positive, careful and constructive assessment.
>
> **Scope.** We deliberately selected Tabula-8B precisely because it is the most openly documented large-scale TLM, with the full T4 training corpus, model weights, and evaluation suite publicly available, enabling the kind of independent verification we conduct here and not possible for less transparent works. UniPredict was also a deliberate choice since it contains datasets with semantically meaningful column names rather than generic identifiers, reflecting realistic real-world tabular settings and is the primary benchmarks in the Tabula-8B evaluation, which the authors specifically highlight as a setting where their model shows particularly strong performance gains. Our literature review (Appendix G, Table 14) demonstrates that the evaluation gaps we identify, such as absent baselines, no task stratification, no contamination analysis, are universal across 5 major TLM papers. The issues are endemic to evaluation practice, not specific to Tabula-8B. That being said, we do explicitly acknowledge this as this limitation (Section 6) and agree that extending to other TLMs is important future work. Lastly, we also directly acknowledge the exclusion of imputation, generation, and multi-table reasoning as a justified limitation since this work was proof-of-principle for deeper systemic flaws in current TLM evaluation protocols. Indeed, these other tasks are crucial next steps for future directions and might have other specific issues/biases that would exist in our prediction efficacy focused evaluation.
>
> **Contamination search scope.** As reviewer j5Lb notes, our targeted search confirms presence but cannot bound absence. We emphasize that contamination was found in 8/10 top-performing dataset and 11 of the top-20 being likely contaminated stock data. We also quantified the deleterious effect of removing these documented cases is Table 12 (and indirectly in Table 5). And we clearly state this as a limitation in Sec. B.3 (L.898). We also kindly ask you to refer to our response to Reviewer j5Lb (Q1) for extended discussion of contamination extent.
>
> We hope this addresses the reviewer's concerns and find this paper to be a valuable contribution to the current literature. We would also appreciate their reconsideration of their overall recommendation. And as always we're happy to address any additional questions or concerns!

---

> > ### Author Rebuttal · Reviewer_3TbE · 2026-04-01
> >
> > I appreciate the authors' good rebuttal. Although some limitations remain, the paper's insights into systemic evaluation flaws make it a valuable contribution worthy of a Weak Accept.

---

> > > ### Author Response · Authors · 2026-04-06
> > >
> > > We thank the reviewer for the constructive assessment and for confirming that our rebuttal addressed their concerns. We will incorporate all discussion points from the rebuttal period into the camera-ready version. Again, thanks so much for your meaningful feedback and helping improve this manuscript!

---

### Official Review · Reviewer_j5Lb · 2026-03-12

**Soundness:** 3
**Presentation:** 3
**Significance:** 4
**Originality:** 3
**Overall Recommendation:** 5
**Confidence:** 4

**Summary:**

The paper re-evaluates Tabula8B on 165 UniPredict datasets and argues that its reported gains are largely artifacts of evaluation rather than evidence of genuine tabular generalization

It shows that strong aggregate results are driven mainly by quartile-binned regression tasks, while binary & categorical classification often provide little or no lift over a majority-class baseline. It also presents evidence that several top-performing datasets are contaminated, including exact train-test overlap, label exposure, and task-level leakage that standard dedup would miss. Finally, the authors show that a non-tabular, instruction-tuned Llama baseline recovers most of Tabula8B’s classification performance, and that simple, quartile-format exposure closes much of the remaining gap, supporting the claim that instruction-following and format familiarity explain much of the benchmark success.

The paper concludes with recommendations for stronger eval standards for tabular language models

**Compliance With Llm Reviewing Policy:**

Affirmed.

**Final Justification:**

Solid diagnostic paper that challenges how tabular LLMs are evaluated. The empirical analysis is careful and directly tests alternative explanations for reported gains.

- Soundness -> strong. The methods are appropriate, and the rebuttal clarifies key details and strengthens confidence
- Significance -> high. The findings have clear implications for evaluation practice.
- Originality -> Moderate... but meaningful. A useful synthesis of issues applied concretely.
- Clarity -> Good, with further improvements in the rebuttal

My main concerns (contamination scope, instruction vs. format effects, reproducibility) were addressed. The contamination analysis is still not exhaustive, but the added evidence is sufficient. The core conclusions remain well supported

The rebuttal reinforced my original assessment.

**Key Questions For Authors:**

**1. How widespread is the contamination?**
 You show several clear examples, but how many UniPredict datasets are affected overall? Also, what happens to the aggregate results if those datasets are removed?

**2. Instruction vs format effects.**
 The Llama experiments suggest format familiarity explains a lot of the performance. Did you try isolating this more directly, e.g., keeping format fixed while varying instruction tuning?

**3.Baseline setup details.**
 Could you clarify the exact prompting setup for the Llama baseline (template, #shots, decoding settings)? This would help reproducibility.

**4.Metric sensitivity.**
 Since quartile regression seems to dominate the benchmark, did you evaluate results under a more balanced task weighting?


**5. Benchmark design guidance**
 Based on your findings, what concrete steps would you recommend for building more reliable tabular LLM benchmarks going forward?

**Limitations:**

Yes.

**Strengths And Weaknesses:**

**Soundness.**
Overall technically solid. The paper runs a series of focused diagnostic experiments that directly test the main hypothesis: that the strong Tabula 8B results come from evaluation artifacts rather than true tabular generalization. The breakdown by task type and the majority-class baseline comparison are sensible and reveal an important issue with aggregate metrics. The contamination checks (train/test overlap, label exposure, task-level leakage) are also thoughtful and go beyond simple deduplication

A couple claims feel a bit stronger than the evidence. The Llama baseline suggests that instruction following and format familiarity explain much of the gain, but those factors are not fully isolated from other differences (prompting, tokenization, etc)
Also, while the paper shows specific contamination examples, it is less clear how widespread these problems are across the full benchmark. Quantifying that would make the argument stronger

**Presentation.**
Generally clear and well organized. The narrative is easy to follow and the main story builds logically from observation -> diagnostic experiments -> interpretation. The figures explaining task-type effects are helpful

Some details could be clearer. In particular, the Llama baseline setup could use a bit more experimental detail (prompt format, shots, evaluation setup). A short early overview of the contamination findings might also help readers orient themselves before the later sections

**Significance.**
The paper tackles an important evaluation issue. If tabular LLM benchmarks can be strongly influenced by task formatting or dataset artifacts, then many reported gains may be misleading. That matters for both researchers and practitioners who are trying to assess whether these models actually generalize to tabular reasoning tasks.

The work is mostly diagnostic rather than methodological, but careful benchmark auditing like this often has real impact. At minimum it should encourage more careful evaluation design for tabular LLMs

**Originality.**
The contribution is analytical: a systematic re-evaluation of an existing benchmark and model. The combination of task-type decomposition, leakage analysis, and baseline comparison provides a new perspective on prior results
Conceptually this builds on existing ideas from benchmark auditing and data contamination analysis, so the novelty is moderate. Still, the insights are useful and likely to stimulate follow-up work

---

> ### Author Rebuttal · Authors · 2026-03-30
>
> We thank the reviewer for the encouraging and thorough assessment. We address the reviewers' remaining minor concerns below.
>
> **Q1: How widespread is contamination?** Our targeted analysis found contamination in 8/10 performing datasets we examined (Table 3). As stated L.898, our analysis represents a *lower bound*. One has to manually search and cross-reference each dataset+task for contamination. Thus, we restricted this search to the top performers where contamination is most consequential. It is not possible for us to manually search the entire corpus. And as our results show, there may be in-direct forms of contamination that evade algorithmic deduplication. Approaches tackling this issue is a crucial and urgent future work direction.
>
> Table12 shows the effect of removing documented cases of contamination. The instruction-tuning contribution rises from 69% to 73%, and the mean lift gap between Alpaca and Tabula shrinks from 4.9 to 4.1 points. On non-stock quartile datasets, the gap disappears entirely (Table 5). The pattern is consistent, the more we can control for contamination, the more Tabula advantage diminishes.
>
> **Q2: Isolating instruction vs format effects.** We want to first clarify that the **input serialization is fixed** across Base, Alpaca, and Tabula. That being said, our design already separates instruction vs format learning effects. Base Llama and Alpaca holds the prompt format fixed (same serialization, same few-shot structure; Appendix A.2) while varying only instruction-tuning, isolating instruction-following. Alpaca vs Alpaca+Q holds instruction-tuning constant while adding only quartile-format exposure, isolating prediction task format familiarity. Together, these comparisons decompose the full Base to Tabula gap.
>
> **Q3: Baseline setup details.** All models use greedy decoding (temperature=0), 4-shot prompting with identical serialized rows. The only difference between Tabula and Alpaca evaluation is the prompt template (special tokens vs. instruction template), serialized content is preserved exactly. Full details are in Appendix A.4; prompt examples for all three task types are in Appendix A.2. We also plan to release all code used for this project, upon acceptance.
>
> **Q4: Metric sensitivity.** Task counts are already reasonably balanced: 64 binary, 55 categorical and 46 quartile. The task-stratified results in Figure 1 and the appendix table already give per-type median lifts, which are more informative than any aggregate weighting scheme. Collapsing across task types into a single weighted number is precisely the aggregation problem our paper identifies as misleading. That being said, Table 13 reports the per-dataset (with task identity) performance metrics for all models, so any alternative task weighting scheme can be tested post-hoc by the community.
>
> **Q5: Benchmark design guidance.** Our seven recommendations in Section 5 directly address this. The most actionable for benchmark builders are: (R1) always report majority-class baselines and Cohen's Kappa alongside raw accuracy; (R2) stratify results by task type and test for heterogeneity; (R4) go beyond row-level deduplication and exclude datasets from widely-replicated sources absent evidence of decontamination; and (R5) verify that top-performing tasks cannot be solved by factual recall or numeric shortcuts from input features alone. It may also be worth assembling a group that maintains private benchmarking data that have never appeared on the open internet to enable contamination-free evaluation.
>
> We thank the reviewer again for their constructive and thoughtful feedback!

---

> > ### Author Rebuttal · Reviewer_j5Lb · 2026-04-03
> >
> > This addressed my main concerns:
> >
> > - The contamination analysis is clearer - showing the effect of removing affected datasets helps. It’s still a lower bound, but directionally consistent with the claim
> > - The instruction vs format decomposition now makes sense & the comparisons are clean and support the interpretation.
> > - Baseline details are sufficient for reproducibility
> > - The stance on stratified metrics vs aggregation is appropriate and consistent with the paper’s thesis
> > - The recommendations are concrete and useful
> >
> > Some limitations remain: (1) The contamination audit isn’t exhaustive, and (2) parts of the mechanism are still somewhat inferential - but these don’t undermine the core result
> >
> > The paper’s main point holds: much of the reported gain comes from task mix, format familiarity, and contamination rather than clear tabular generalization

---

> > > ### Author Response · Authors · 2026-04-06
> > >
> > > We sincerely thank the reviewer for their continued thoughtful engagement and for maintaining their positive assessment! This has genuinely helped us improve the manuscript. We particularly appreciate the reviewer's precise characterization of remaining limitations, specifically the contamination audit being a lower bound, which we agree with and will ensure are clearly stated in the final version.
> > >
> > > We also want to highlight the reviewer's closing summary, which we believe captures the paper's thesis well: the reported gains are substantially explained by task mix, format familiarity, and contamination rather than clear tabular generalization. Our goal is for this finding, along with the concrete recommendations in Section 5, to serve as a foundation for more rigorous TLM evaluation going forward.

---

### Official Review · Reviewer_QdZo · 2026-03-13

**Soundness:** 3
**Presentation:** 3
**Significance:** 2
**Originality:** 2
**Overall Recommendation:** 4
**Confidence:** 4

**Summary:**

The paper does an extensive study on various limitations associated with evaluation of Tabular Language models (TLMs) - data contamination, stratification of different types of tasks to a single number (quartile tasks vs binary tasks), lack of majority baselines, importance of instruction tuning for the high performance/generalization of TLMs and thus calls for potential recommendations/actions.

**Compliance With Llm Reviewing Policy:**

Affirmed.

**Final Justification:**

The authors have addressed most of my concerns during the rebuttal and once the paper is updated with all the rebuttal discussions, I believe it will be at much better shape. So, I've updated my final recommendation accordingly.

**Key Questions For Authors:**

- [Typo] L408, right: Feasible

**Limitations:**

Yes, there's a Limitations and Impact statement section.

**Strengths And Weaknesses:**

**Strengths**

- Very well written, clear motivation and call for broader community, practitioners and researchers in TLM community on evaluation (refer weaknessess for possible extensions and strengthening the critique).
- I like the overall stance of the paper along with the recommendations provided.

**Weaknessess**

For the experiments, make necessary assumptions.
- [**Discussion, Experiment, Confounding of instruction-tuning vs tabular learning; Sec 4.3**] Pretrained (Llama-base) -> Instruction-tuned (Alpaca) led to a massive jump while instruction-tuned -> tabular-tuned (Tabula) led to a decent jump. An experiment to consider is to pick a pretrained model and just do tabular tuning (i.e Llama-base -> Tabula) which will give more insights on this claim i.e by skipping instruction-tuning. I believe training base models for tabular tasks can be done by borrowing some inspiration from tabular synthesis methods [1, 2, 3].
- [**Discussion, Experiment, Data leakage; Sec 4.2**] An experiment to tease out the factors of data leakage on performance of the tabular performance will strengthen the claim even further as the current summary (Table 4) can be viewed as correlation rather than causation. One way I can think of is: Unlearn those data points; so those were "technically" not seen by model during training and then measure the performance.
- [**Discussion, Extension of survey**] I believe Appendix G can be extended as the authors mention it as "survey" (L401-411, Sec 6). At the minimum, add a long table evaluating most recent/famous papers in TLM with your framework. This will help practitioners, reviewers and broader community to quickly understand where to focus on evaluations and which papers lack in specific dimensions.
    - [**Discussion, Evaluation framework on close source TLMs**] I encourage the authors to discuss how to apply the framework for any close-source TLMs as it's very difficult to evaluate them without knowing the details.
- [**Discussion, Extension of framework analysis/critique for different TLM tasks**] In TLM, another area worth critiquing is the use of these models for table synthesis [1,2,3] and how they are fairing under this framework where applicable. This can go in the long table as part of extending the survey and/or a separate small section in Appendix as TLM community will be benefitted with broader evaluation guidelines.

1. Tabby: Tabular Data Synthesis with Language Models: https://arxiv.org/abs/2503.02152
2. Language Models are Realistic Tabular Data Generators: https://arxiv.org/abs/2210.06280
3. Generative Table Pre-training Empowers Models for Tabular Prediction: https://arxiv.org/abs/2305.09696

---

> ### Author Rebuttal · Authors · 2026-03-30
>
> We thank the reviewer for their constructive engagement and specific suggestions. We address each concern below.
>
> **Llama-base → Tabula ablation already exists in our results.** We really appreciate this suggestion, and want to clarify that this experiment already exists in our evaluation. Tabula-8B is fine-tuned directly from __base__ Llama-3-8B on the __seralized__ T4 tabular corpus (Gardner et al., 2024). Alpaca is a separate branch from the same base, fine-tuned only on instruction data. The two are parallel paths:
>
> Base Llama → tabular data (T4) → Tabula-8B (tabular pretraining only, no instruction-tuning)
> Base Llama → instruction data → Alpaca (instruction-tuning only, no tabular data)
>
> Our decomposition (Table 4) shows that Base Llama (47.7%) → Alpaca (58.6%) vs. Tabula-8B (63.5%). Since both Alpaca and Tabula start from the same base, the comparison reveals that instruction-tuning alone (Alpaca) recovers 92.2% of Tabula's accuracy without any tabular exposure. And that the gain from T4-specific training is marginal and much of which can be attributable to contaminated datasets.
>
> To put it differently, if tabular pretraining were the primary driver, we would expect Tabula-8B (which has T4 exposure) to substantially outperform Alpaca (which has none). Instead, the gap is small, much of which traces to contaminated datasets (see below). Hence, the reviewer's proposed experiment already exists in our evaluation and yields the conclusion they predicted it would test.
>
> We will clarify this parallel structure more explicitly in Section 3 (Approach) to avoid future ambiguity.
>
> **Contamination evidence is structural, not correlational.** We clarify that our contamination evidence is direct and structural. We found 753/753 test rows with exposed labels in T4 (us-womens-labor), target values in multiple T4 chunks for all examined financial datasets, and 844 task-level matches for date-day associations for a single test example (Peloton). This is not a statistical association requiring causal intervention, it is a documented presence of evaluation data (with labels) in the training corpus. Moreover, Table 5 provides a natural experiment where partitioning quartile datasets by contamination risk reveals that Tabula's advantage exists only on stock/financial datasets where we documented direct label exposure. On non-stock datasets, Alpaca+Q matches Tabula. This is the pattern the reviewer's unlearning experiment would test, and it already points clearly in one direction.
>
> That being said, we had not considered unlearning as a framework for addressing this problem, and we thank the reviewer for this suggestion. It could be a genuinely interesting method’s framework to tackle the issues our work uncovers. However, machine unlearning for LLMs at this scale (T4 ~100B tokens) remains an active research area with known reliability limitations, and developing a robust unlearning method here would constitute a substantial methodological contribution in its own right, which would be beyond the scope of this evaluation paper. Though, we will certainly discuss unlearning as a promising future direction.
>
> **Survey extension, closed-source TLMs, and table synthesis.** These are valuable suggestions. We will expand Appendix G into a structured evaluation table including synthesis methods (Tabby, GReaT, TapTap) in the camera-ready version. We will also add discussion of closed-source TLM evaluation challenges and table synthesis evaluation. We note that our work’s recommendations are applicable to closed-source TLMs too and instruction-tuned controls do not require direct model access.
>
> **Typo.** Fixed, thank you!
>
> We thank the reviewer again for their constructive feedback! And if these responses have sufficiently addressed the reviewer’s concerns, we would appreciate a reconsideration of the overall recommendation.

---

> > ### Author Rebuttal · Reviewer_QdZo · 2026-04-02
> >
> > Thank you so much for the detailed rebuttal.
> >
> > Regd **Llama-base → Tabula ablation**: Thanks a lot for correcting my understanding. Please update the section to avoid ambiguity.
> >
> > Regd **Contamination evidence**: I understand and agree the limitations of bringing up unlearning frameworks to the study and appreciate the authors for considering expanding the paper to include this as a potential future direction. While it's anecdotal, I am still not entirely convinced with *verbatim being equal to contamination*, I agree that Table 5 serves as a natural experiment for unlearning.
> >
> > Regd **Survey extension, closed-source TLMs, and table synthesis.**: Appreciate the extensions.
> >
> > Overall, my questions have been mostly answered and I've updated my score to reflect the same. Please update your final version with all the rebuttal discussions (incl. other reviewers who raised some good questions) and all the very best with your submission.

---

> > > ### Author Response · Authors · 2026-04-06
> > >
> > > We thank the reviewer for raising their score and for their rigorous, constructive engagement throughout this process!
> > >
> > > We will update Section 3 to make the parallel decomposition structure unambiguous, as the reviewer's original question revealed this was not sufficiently clear. We will also expand Appendix G to cover synthesis methods and add discussion of closed-source TLM evaluation, as promised.
> > >
> > > We appreciate the reviewer's candor on the contamination point. What our evidence does establish is that (a) evaluation data with labels exists in training data, and (b) the performance advantage disappears precisely on datasets where contamination risk is controlled. We believe this combination is sufficient to warrant the caution our paper recommends.
> > >
> > > We thank the reviewer again for the suggestions, several of which have directly improved the paper.

---

### Official Review · Reviewer_kGsm · 2026-03-13

**Soundness:** 3
**Presentation:** 1
**Significance:** 2
**Originality:** 2
**Overall Recommendation:** 4
**Confidence:** 3

**Summary:**

This submission re-evaluates TLMs and chooses Tabula-8B as an example, and challenges claims of emergent tabular generalization.

The authors reveal that reported high performance is actually driven by task-type bias, pervasive data contamination, and general instruction-following capabilities rather than genuine tabular reasoning.

The authors conducted the experiments by LoRA-SFT LLama-8B with alpaca, and continue training on following experiments.

The authors have formulated many recommendations for TLM evaluation, while the authors did not conduct them.

**Compliance With Llm Reviewing Policy:**

Affirmed.

**Final Justification:**

The rebuttal has addressed most parts of my concerns and I have raised my recommendations accordingly.

**Key Questions For Authors:**

1. I appreciate the authors courage to challenge the previous research results and paradigms. But for a not-closely engaged reader, I am very confused of the submission's motivation, research question analytical framework. The the central question 'Do TLM truly generalize to tabular data' by author is not sufficiently formulated.

2. Vspaces are adjusted in

   1. Figure 3
   2. before section 4 Findings

**Limitations:**

yes

**Strengths And Weaknesses:**

### Strength

1. Exposed the systemic flaws in current TLM evaluations: data contamination, leakage, etc.
2. disentangled 'instruction-following' capabilities from 'tabular reasoning
3. The experiment results are extensive, showing the flaws in TLM evaluation is questionable.
4. This work shall be an position paper for TLM evaluation, and inspire following researchers to avoid the exposed flaws.

### Weakness

1. No explicit contribution. Though the authors have summarized in the end of introduction, the authors did not present any technical design, or validate their that is typically required for a conference paper.
2. The writing is not well-polished to accommadate to a reader-friendly state. The intents of the authors within paragraphs are implicit.
3. The writing and organization is not typical for ICML. I am not sure whether this submission is distributed to a wrong track or to me. Current form (more like a blog post) is not well-motivated and thus not suited for submission/publication.
   3.1. The authors shall introduce how they define the tabular prediction task, or at least present a clear motivation based on these previous works. Currently I am confused and fail to recognize the authors attitutes/statements towards the listed points.
   3.2. Section 3: rather than approach, this section is actually telling the experiment setting. The motivation behind this is absent: can the authors conclude an overall structure of the analysis process?
   3.3. Disclaimer section after section 3: why we need this section in the paper, especially the bolded ones?
   3.4. Section 5: these recommendations are presented as not being based on concrete/direct experiment results. The authors shall link each of the recommendations to the previous findings and explain the logics.
4. The findings/analysis are largely based on how the authors conduct the experiment (SFT on alpaca and etc.). However, the authors have adopt the LoRA finetuning, which is not the standard practice. Will the analysis results be generalizable to full-parameter SFT setting?
5. The authors shall at least try one of the proposed recommendations. Currently this paper only exposed the gaps yet not fixing it.

---

> ### Author Rebuttal · Authors · 2026-03-30
>
> We thank the reviewer for their time to review our work. We address each concern below.
>
> **Evaluation papers are in-scope at ICML.** The ICML 2026 Call for Papers explicitly lists "evaluation (methodology, meta studies, replicability and validity)" as a topic of interest. Our work is a systematic study of TLM evaluation validity and shortfalls. Thus, we believe our paper is of interest and appropriate for this venue.
>
> **Central research question formulation.** We operationalize "Do TLMs generalize?" as three falsifiable sub-questions: (1) Does TLM performance exceed naive baselines across task types? (2) Can top performance be attributed to training data contamination? (3) Is tabular pretraining necessary, or does general instruction-following suffice? Each is addressed by a distinct experimental finding.
>
> **We implemented 6 of 7 proposed recommendations as our paper.** The reviewer states we did not conduct our recommendations. We clarify that the paper's experimental design is the implementation.
> - R1 (baselines + Cohen's κ): Tables 1, 6-7, Figure 5.
> - R2 (task-type stratification): Figures 1-2, Tables 7-8, ANOVA.
> - R4 (contamination beyond row-level dedup): Section 4.2, Tables 2-3, 9-10, Figures 3, 6-7.
> - R5 (task auditing): Appendix D.
> - R6 (instruction-tuned baselines): Section 4.3, Tables 4-5, 11-12, Figure 4.
> - R7 (community guidance): Appendix G.
>
> The only recommendation not implemented is R3 (release of evaluation code by TLM authors), which requires action from original authors, not us. We instead report the raw performance metrics for each of the 4 models across 165 datasets evaluated and plan to release all evaluation code, upon acceptance. Our recommendations emerge directly from, and are validated by, our experiments.
>
> **LoRA matches full fine-tuning on instruction-tuning tasks; the choice of fine-tuning method does not affect our conclusions.** We used LoRA fine-tuning (Hu et al., 2022) due to compute constraints, while Tabula-8B uses full fine-tuning. However, LoRA has been shown to match full fine-tuning performance on small-to-medium instruction-tuning datasets (Dettmers et al., "QLoRA: Efficient Finetuning of Quantized LLMs", NeurIPS 2023; Schulman et al., "LoRA Without Regret", Thinking Machines Lab, 2025). The difference in fine-tuning approach is therefore unlikely to explain the 92.2% recovery, and does not change the directional conclusion that instruction-following alone accounts for the large majority of Tabula-8B's performance.
>
>
> **On presentation.** We are more than happy to integrate any specific and constructive suggestions the reviewer might have to improve readability. Here’re specific changes we plan on making in the final version based on your comments:
> - Re 3.2: Sec. 3 describes the experimental design (baselines, evaluation scope, contamination search methodology) and "Approach" is a standard term for this in evaluation/benchmarking papers. We will update Sect. 3 to open with an explicit statement of the analytical framework.
> - Re 3.3: We will reposition the Disclaimer to improve flow of the paper; note that we felt this was necessary because critical papers risk being misread as personal attacks.
> - Re 3.4: More explicitly linking each recommendation to our experiments/results
> - Fix any spacing issues
>
> **Minor clarification.** Re 3.1: The tabular prediction task is defined in Section 2 (Background) and task definitions follow the Tabula-8B protocol exactly (Gardner et al., 2024), with quartile classification defined in the first para of Sec 4.1.2
>
> In summary, evaluation papers are explicitly in-scope at ICML, our recommendations are implemented and validated by our experiments, and LoRA provides a valid estimate of instruction-tuning's contribution.
>
> We hope our responses have addressed all of the reviewer's concerns and demonstrated the paper's contribution to current literature. If so, we would appreciate a reconsideration of the overall recommendation.

---

> > ### Author Rebuttal · Reviewer_kGsm · 2026-04-03
> >
> > The authors rebuttal have clarified my understandings to the submission. I will apology for the previous misunderstandings. After consideration, I decided to raise my recommendation.
> >
> > The author can choose not to reply the following concern:
> >
> > I currently regard that the submission is making a wide-covered claim for TLM, yet not covering a full SFT experiment is blocking my further recommendation, since most following analysis are based on the setting.
> >
> > There are literatures showing LoRA's directional effectiveness. But for fair comparison, I maintian my suggestion that the authors perform the full SFT setting while I understand the computing budget issue. the neglection on this issue potentially risks misleading the community.

---

> > > ### Author Response · Authors · 2026-04-06
> > >
> > > We sincerely thank the reviewer for raising their assessment and for constructively engaging with our rebuttal!
> > >
> > > We would like to gently highlight two points on full SFT. First, as referenced above, LoRA has been shown to match full fine-tuning on instruction-tuning tasks of this scale, so the 92.2% recovery (and direction) is unlikely to be an artifact of the fine-tuning method. Second, and more importantly, the primary risk with LoRA is that it *under-tunes relative to full SFT*. If full SFT yields a meaningful difference from LoRA, it would most likely result in Alpaca recovering *more* of Tabula-8B's performance, which would *strengthen our central finding* that instruction-following is a key driver of performance. The directional conclusion is robust to this choice, and we believe there is no risk of misleading the community on this point.
> > >
> > > We appreciate the reviewer's continued engagement and feel this exchange has improved the paper.

---

### Decision · Program_Chairs · 2026-04-30

**Decision:**

Accept (regular)

**Comment:**

This submission presents a systematic re-evaluation of tabular language model performance, focusing on Tabula-8B and the UniPredict benchmark. The paper argues that much of the apparent performance is explained by evaluation artifacts, including class-imbalance effects, contamination, and instruction-following or format familiarity rather than true tabular reasoning. Reviewers found the paper timely and important, and several explicitly viewed it as a valuable corrective contribution for a fast-moving area.

The strongest aspect of the paper is the diagnostic structure of the evidence. The breakdown by task type, the explicit majority-class comparisons, the contamination analysis beyond simple deduplication, and the instruction-tuning comparisons all point in the same direction. The main limitations raised by reviewers were that the study is centered on a representative case study rather than the entire field, and that some of the broadest claims should be calibrated accordingly.

Overall, I recommend accept. The reviews are broadly positive, but they ask the authors to keep the claims carefully calibrated and make explicit which conclusions are directly established for Tabula/UniPredict versus broader implications for TLM evaluation.